# Efflux pump gene amplifications bypass necessity of multiple target mutations for resistance against dual-targeting antibiotic

Kalinga Pavan T. Silva[1], Ganesh Sundar[1] & Anupama Khare ®[1]✉

Antibiotics that have multiple cellular targets theoretically reduce the frequency of resistance evolution, but adaptive trajectories and resistance mechanisms against such antibiotics are understudied. Here we investigate these in methicillin resistant *Staphylococcus aureus* (MRSA) using experimental evolution upon exposure to delafloxacin (DLX), a novel fluoroquinolone that targets both DNA gyrase and topoisomerase IV. We show that selection for coding sequence mutations and genomic amplifications of the gene encoding a poorly characterized efflux pump, SdrM, leads to high DLX resistance, circumventing the requirement for mutations in both target enzymes. In the evolved populations, *sdrM* overexpression due to genomic amplifications containing *sdrM* and two adjacent genes encoding efflux pumps results in high DLX resistance, while the adjacent hitchhiking efflux pumps contribute to streptomycin cross-resistance. Further, lack of *sdrM* necessitates mutations in both target enzymes to evolve DLX resistance, and *sdrM* thus increases the frequency of resistance evolution. Finally, *sdrM* mutations and amplifications are similarly selected in two diverse clinical isolates, indicating the generality of this DLX resistance mechanism. Our study highlights that instead of reduced rates of resistance, evolution of resistance to multi-targeting antibiotics can involve alternate high-frequency evolutionary paths, that may cause unexpected alterations of the fitness landscape, including antibiotic cross-resistance.

The emergence of antimicrobial resistance (AMR) is a major threat to global health. A recent report indicated that there may have been more than one million deaths worldwide due to AMR in 2019[1]. Methicillin-resistant *Staphylococcus aureus* (MRSA) causes a wide-range of healthcare- and community-associated infections, with high incidence and mortality rates, and can attain resistance against most available antibiotics[2–7]. Resistance determinants in bacteria are typically acquired either via horizontal gene transfer or de novo mutations, and mechanisms include antibiotic degradation or sequestration, modification of target components, and overproduction of efflux pumps[8–11]. Additionally, ubiquitous, and unstable genomic duplications and amplifications can lead to increased expression of modifying

enzymes and efflux pumps that then confer antibiotic resistance and may be selected for upon antibiotic exposure[12–14].

One strategy that has been proposed to reduce the rise of antibiotic resistance is to develop antibiotics with more than one target, thereby reducing the frequency of resistance evolution[15,16]. Recent studies have identified two such dual-targeting antibiotics that target membrane integrity as well as an additional cellular pathway and have so far avoided the emergence of resistance in the laboratory[17–19]. The bacterial topoisomerases, DNA gyrase, and topoisomerase IV, have also been proposed as potential targets for dual-targeting antibiotics, and it has been shown that targeting both enzymes may inhibit resistance evolution, and potentially involve novel resistance

[1]Laboratory of Molecular Biology, National Cancer Institute, National Institutes of Health, Bethesda, MD 20892, USA. ✉e-mail: anupama.khare@nih.gov

determinants[15,16,20,21]. However, the mechanisms of resistance evolution to such multi-targeting antibiotics and the underlying adaptive trajectories have not been characterized.

Fluoroquinolones are a widely-used class of antibiotics, and most traditional fluoroquinolones, such as ciprofloxacin and levofloxacin, preferentially target either DNA gyrase or topoisomerase IV[22]. Delafloxacin (DLX) is a 4th generation fluoroquinolone antibiotic, which targets both the DNA gyrase and topoisomerase IV enzymes with similar potency[23–25]. Due to this dual-targeting, it was thought that resistance against DLX might be infrequent[24,26,27], but DLX resistance has recently been observed in clinical isolates of *S. aureus*[28,29].

In this study we investigate the evolution of MRSA resistance to dual-targeting antibiotics, using the experimental evolution of multiple independent MRSA populations in increasing concentrations of DLX. We observe that in addition to mutations in the DNA gyrase and topoisomerase IV enzymes, coding sequence mutations in the major facilitator superfamily efflux pump SdrM (*Staphylococcus* drug resistance), and genomic amplifications of *sdrM* and the neighboring efflux pumps *sepA* and *lmrS* are widespread in the evolved populations, and typically evolve earlier than the canonical mutations. The *sdrM* coding sequence mutations confer moderate DLX resistance, and increase evolvability of such resistance, while the genomic amplifications lead to high DLX resistance. Copy number variation of the amplified region is dependent on the selective pressure and causes population heterogeneity for DLX resistance. We find that while *sdrM* activity provides the fitness advantage for selection of these genomic amplifications upon DLX exposure, hitchhiking of the neighboring efflux pumps in the genomic amplification leads to cross-resistance against the aminoglycoside streptomycin. Additionally, we show that in the absence of *sdrM* activity, DLX resistance requires mutations in both the DNA gyrase and topoisomerase IV enzymes, and thus arises at a lower frequency. Finally, we demonstrate that *sdrM* genomic amplifications and mutations are also common in populations from one MRSA and one methicillin-sensitive *S. aureus* (MSSA) clinical isolate each upon selection for DLX resistance.

## Results

### Multiple evolved MRSA populations likely have novel determinants of DLX-resistance

We evolved ten independent populations of the MRSA strain JE2 in increasing concentrations of DLX for 7–10 daily passages. DLX inhibits both the DNA gyrase and topoisomerase IV enzymes with similar potency raising the possibility that resistance to DLX may develop

infrequently[27]. However, all ten populations were able to grow in DLX concentrations that were 64–1024× the minimum inhibitory concentration (MIC) of the parental JE2 strain (Supplementary Dataset 1, see Materials and Methods for strain nomenclature). The JE2 strain had a MIC of 0.133 ± 0.027 μg/ml in MH2 medium, which is below the clinical breakpoint of 0.25 μg/ml[23–25]. Three individual isolates from the terminal passage of each evolved population were tested for DLX resistance and all isolates showed DLX MICs ranging between ~2 and 33 μg/ml (Supplementary Fig. 1a), confirming the evolution of high DLX resistance. We carried out whole-genome sequencing on these isolates, as well as additional isolates and populations from earlier passages of the evolution (Supplementary Dataset 2). As expected, several resistant isolates and populations had mutations in genes encoding the subunits of either DNA gyrase (*gyrA* or *gyrB*) or topoisomerase IV (*parC* or *parE*) or both (Fig. 1). Most of the single nucleotide polymorphisms (SNPs) that arose in these canonical targets such as S85P and E88K in *gyrA*, R458L in *gyrB*, E84K and A116V in *parC*, and D432G and P585S in *parE* are located in the quinolone resistance determining region (QRDR) of these proteins and have been previously associated with fluoroquinolone resistance[26,30–33]. We also identified other mutations such as W592R in *gyrB*, S520R in *parC*, and S437P in *parE* in our mutants, which may be novel DLX resistance alleles.

Despite the numerous canonical target mutations, several resistant isolates as well as populations from intermediate passages of most of the ten independent populations did not carry any canonical mutations in genes coding for either the DNA gyrase or topoisomerase IV enzymes or had mutations in only one of the targets (Fig. 1 and Supplementary Dataset 2). These include all samples from populations 1, 3, and 6, as well as the populations from multiple early passages of populations 2, 4, 5, 7, 8, and 9. Evolved *gyrA** (*gyrA^{E88K}*), or *parE** (*parE^{D432G}*) mutant alleles individually led to a mild increase in DLX MICs up to 0.4 μg/ml (Supplementary Fig. 1b) suggesting that other genes likely play a role in DLX resistance in several of our evolved populations.

### Efflux pump mutations and gene amplifications were prevalent in the evolved mutants

Further examination of our whole-genome sequencing results revealed that mutations in the efflux pump SdrM (*Staphylococcus* drug resistance) were common in the evolved populations (Fig. 1). SdrM is an efflux pump from the major facilitator superfamily (MFS)[34]. Overexpression of SdrM has been shown to confer a 2-fold increase in norfloxacin and ethidium bromide MICs, however, this efflux pump has not been implicated in the evolution of antibiotic resistance[34].

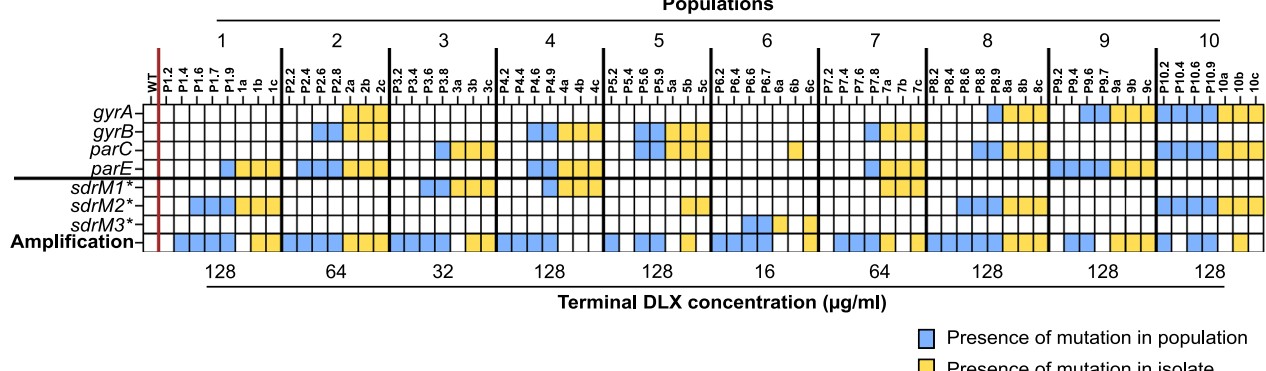

**Fig. 1 | Efflux pump gene amplifications and *sdrM* polymorphisms are widespread in independently evolved populations.** The presence of mutations in genes encoding DNA gyrase subunits (*gyrA*, *gyrB*) and DNA topoisomerase IV subunits (*parC*, *parE*), the three mutant alleles *sdrM1**, *sdrM2**, and *sdrM3**, and a genomic amplification containing *sdrM*, are shown for populations from intermediate passages, as well as the three isolates from the terminal passage, for the ten

independently evolved populations. For each independent population, earlier to later passages are shown from left to right. Blue or yellow squares show the presence of the mutation (SNP or amplification) in a population or a terminal isolate, respectively. The terminal DLX concentrations represent the final concentrations of the evolution experiment (at which point the isolates were obtained).

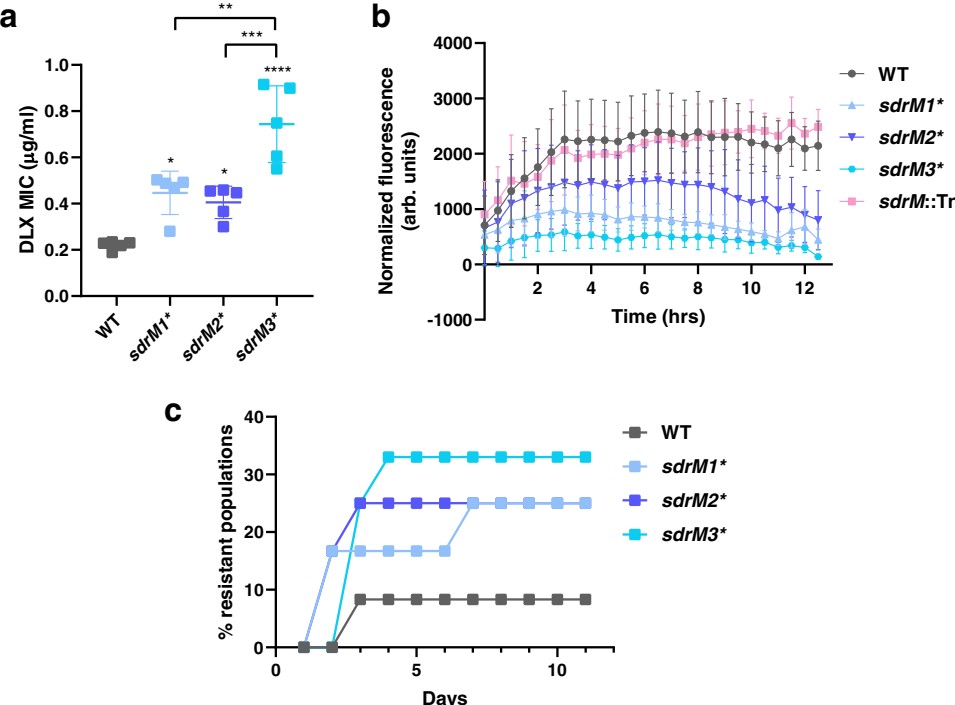

**Fig. 2 | Evolved *sdrM* alleles increase DLX resistance and evolvability. a** DLX MICs of WT, and the three *sdrM* allelic replacement mutants in M63. Data shown are the mean ± standard deviation of five biological replicates. Significance is shown for comparison to the WT, or between mutants as indicated, as tested by a one-way ANOVA with Tukey's test for multiple comparisons (*$P < 0.05$, $P$ for WT vs *sdrM1* = 0.0138, WT vs *sdrM2* = 0.0486, **$P < 0.01$, *sdrM1** vs *sdrM3** $P = 0.0015$, ***$P < 0.001$, *sdrM2** vs *sdrM3** $P = 0.0004$, ****$P < 0.0001$). **b** Normalized fluorescence (intrinsic fluorescence of DLX/$OD_{600}$) was measured for the indicated strains, as a proxy for the intracellular DLX concentration. Data shown are the mean ± standard error of three biological replicates. **c** Percentage of the 12 independent populations of the indicated strains that evolved DLX resistance over time when evolved in 2.5x the respective DLX MICs. Source data are provided in the Source Data file.

Contrary to typical efflux pump mutations that occur in the promoter region (or in a repressor), we observed that eight out of the ten evolved populations contained one of two *sdrM* coding sequence mutations, Y363H (*sdrM1**) or A268S (*sdrM2**), but none of the evolved isolates had both (Fig. 1, Supplementary Dataset 2), raising the possibility that these two mutations might have similar functionality. Population 6 had a mutation upstream of the *sdrM* coding sequence (at the −164 position) which may affect the regulation of *sdrM* expression, in addition to the A268S mutation (*sdrM3**).

Alignment of the SdrM amino acid sequence with other known MFS efflux pumps using the Conserved Domain Database[35] showed that the A268 and Y363 residues are likely situated in the binding pocket of the SdrM efflux pump (Supplementary Fig. 2), suggesting that the mutations affect binding of SdrM to DLX. Visualization of the predicted protein structure of SdrM using AlphaFold[36,37], indicated that A268 and Y363 are in close proximity to each other (Supplementary Fig. 3).

In addition to the *sdrM* alleles, coverage analysis of the whole-genome sequencing results revealed that the coverage of the *sdrM* gene was 2–5-fold higher than the mean coverage of the genome in multiple evolved mutants, compared to ~1× coverage in the WT, indicating an amplification of the *sdrM* gene across all ten evolved populations (Fig. 1, Supplementary Dataset 2).

### *sdrM* mutant alleles increase DLX resistance and efflux, and the evolvability of DLX resistance

We constructed individual *sdrM* allele-replacement mutants in the wild-type (WT) JE2 strain and determined the effect of the evolved alleles on DLX resistance and efflux. Constructed mutants with the *sdrM1** or *sdrM2** alleles showed a ~2-fold increase in DLX resistance, while those carrying the *sdrM3** allele (that consists of both an intergenic and coding sequencing mutation) showed a ~4-fold increase

(Fig. 2a). We measured the intrinsic fluorescence of DLX[38] to indirectly determine the intracellular concentration of DLX (Supplementary Fig. 4a). We calculated the rate of efflux in the three allelic replacement mutants, as well as the WT and a mutant with a transposon insertion in *sdrM* (*sdrM*::Tn) as controls (Fig. 2b). As expected, the WT and *sdrM*::Tn strains had the highest intracellular DLX concentrations, while the three allele-replacement mutants had lower levels, suggesting that the evolved *sdrM* alleles led to increased efflux. Using a simplified mathematical model, we determined that the rates of DLX efflux of *sdrM1**, *sdrM2**, and *sdrM3** were ~2–9x higher than *sdrM*::Tn, while the WT rate was similar to *sdrM*::Tn. (Supplementary Fig. 4b).

From our whole-genome sequencing results, we observed that in eight out of the ten evolved populations *sdrM* mutations (either a SNP or the amplification) emerged at an earlier passage compared to the canonical DNA gyrase or topoisomerase IV mutations (Fig. 1). This suggests an evolutionary cascade in which efflux pump mutations facilitate the selection of canonical mutations for DLX resistance. To test if *sdrM* mutations can affect the evolvability of resistance in the presence of DLX, we evolved 12 independent populations each of WT, and mutants carrying the individual *sdrM* evolved alleles with daily passages in fixed concentrations of delafloxacin, that were 2.5x the respective MICs. While only one WT population evolved resistance, 3–4 populations each of *sdrM1**, *sdrM2**, and *sdrM3** evolved resistance during the experiment (Fig. 2c).

We examined the allelic diversity of the SdrM protein to determine whether the evolved alleles we identified (A268S and Y363H), or any other mutations in the binding pocket of SdrM, are seen in publicly available genomes of *S. aureus* isolates. The JE2 SdrM protein sequence was queried against a set of 63,980 *S. aureus* genomes from the NCBI Pathogen Detection database. While no mutations were seen at position Y363, we identified one strain with an A268S mutation, and a

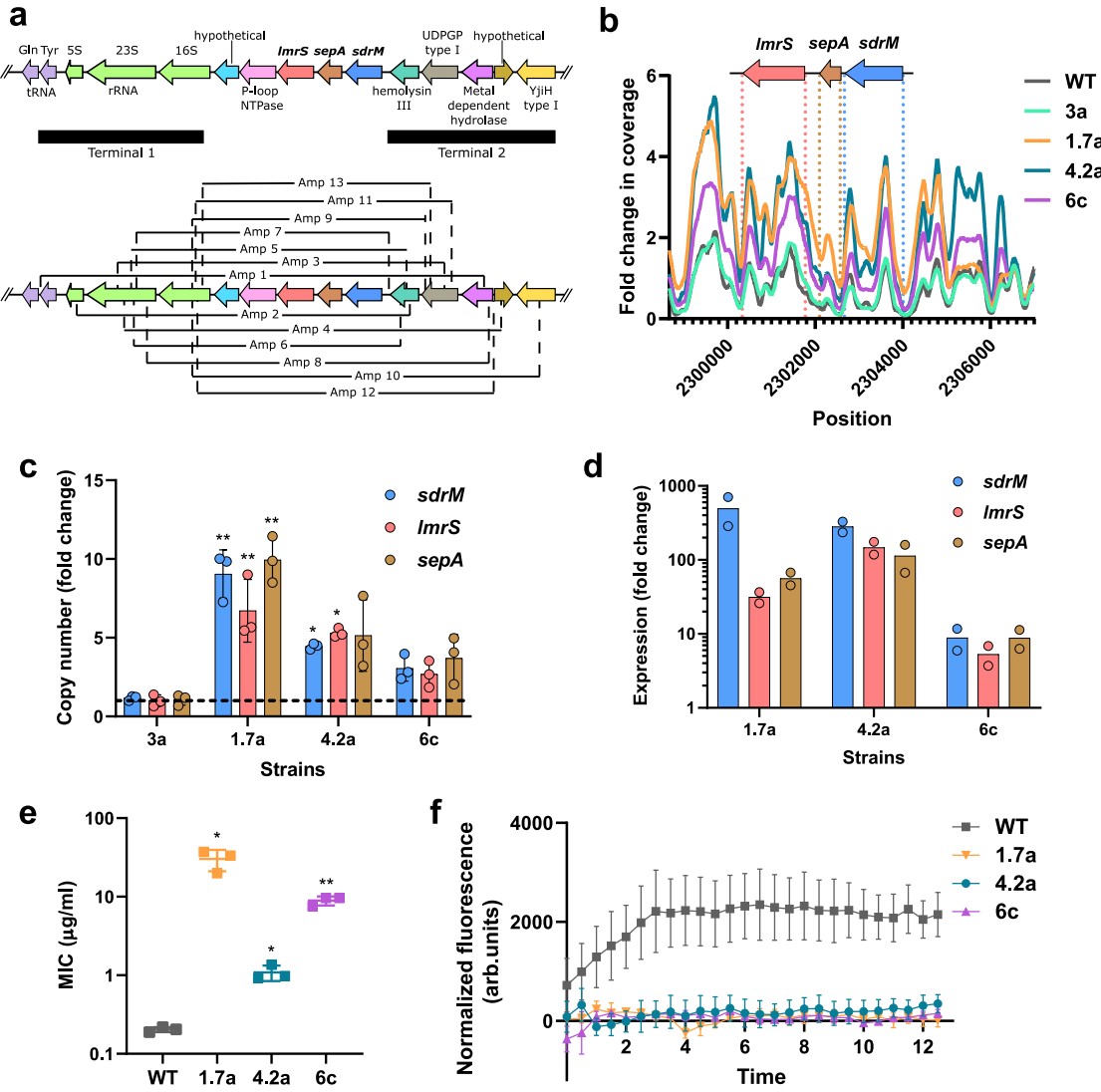

**Fig. 3 | Evolved isolates with gene amplifications have high efflux pump expression and DLX resistance. a** The *sdrM* genomic locus and the 13 distinct types of efflux pump gene amplifications seen in the evolved populations. The regions containing the two ends of the amplification in all cases are shown (Terminals 1 and 2). **b** Relative read coverage of the amplified regions compared to the entire genome for the indicated strains. The lines represent a smoothed fit using a generalized additive model considering the nearest 100 neighbors. **c** Fold-change of the copy number of *sdrM*, *lmrS*, and *sepA* in the indicated isolates compared to the WT as measured by qPCR of genomic DNA. Data shown are the mean ± standard deviation of three technical replicates. **d** Fold-change in the expression of *sdrM*, *lmrS*, and *sepA* in the indicated isolates compared to WT, as measured by RT-qPCR. Data shown are the mean of two biological replicates.

**e** DLX MICs of the indicated isolates in M63. Data shown are the mean ± standard deviation of three biological replicates. **f** Normalized fluorescence (intrinsic fluorescence of DLX/OD$_{600}$) of the indicated strains, as a proxy for the intracellular DLX concentration. Data shown are the mean ± standard error of three biological replicates. Significance is shown for comparison to **c** WT value set to 1 as tested by a Kruskal–Wallis test followed by an uncorrected Dunn's test for each comparison and **e** WT as tested by Brown-Forsythe and Welch one-way ANOVA tests, followed by an unpaired two-tailed *t* test with Welch's correction for each comparison (*$P < 0.05$, **$P < 0.01$, ***$P < 0.001$, ****$P < 0.0001$). **c** 1.7a: $P$ for *sdrM* = 0.0025, *lmrS* = 0.0079, *sepA* = 0.0025, 4.2a: $P$ for *sdrM* = 0.0279, *lmrS* = 0.0438; and **e**, $P$ for 1.7a = 0.0295, 4.2a = 0.0238, 6c = 0.006. Source data for **b**–**f** are provided in the Source Data file.

further seven strains with an A268V mutation (Supplementary Dataset 3). Additionally, these *S. aureus* strains harbored numerous mutations in other residues of the predicted SdrM binding pocket.

**Efflux pump gene amplifications play a role in DLX resistance**
In addition to the point mutations in *sdrM*, we identified 13 distinct types of amplifications from our whole-genome sequencing data across our ten evolved populations, that amplified the *sdrM* gene (Figs. 1 and 3a, Supplementary Dataset 4, Supplementary Fig. 5). At least one instance of each amplification type was confirmed via PCR and Sanger sequencing of the novel junctions. One end of every amplification was within an rRNA-tRNA gene cluster located downstream of *sdrM* (terminal 1), while the other end was either within or

between the five genes upstream of *sdrM* (terminal 2) (Fig. 3a). While five of these amplifications had microhomology of 4–12 base pairs between the two terminals, the others had either a single base pair homology or no homology (Supplementary Dataset 4). Several of the amplifications were found in multiple populations and in some populations, different amplifications were seen in different passages, indicating that the amplifications are dynamic (Supplementary Fig. 6). The *sdrM* gene is present upstream of two additional efflux pumps *sepA* and *lmrS*[39,40], and both these genes were present within the amplifications in all cases (Fig. 3a). Gene neighborhood analysis using the STRING database[41] indicated that the three efflux pumps (along with a hemolysin III family protein) are found next to each other in the genome in almost all *Staphylococcus* species (Supplementary Fig. 7).

To further characterize the effects of the genomic amplifications, we analyzed three evolved isolates that contained only non-canonical mutations – 1.7a, 4.2a, and 6c. All three isolates were predicted to have efflux pump amplifications, where 1.7a had amplification type 9 and both 4.2a and 6c had amplification type 1 (Fig. 3b, Supplementary Dataset 4). While 4.2a had the WT *sdrM* allele, 1.7a had the *sdrM2** allele, and 6c had ~40% reads mapping to the *sdrM3** allele in the whole-genome sequencing data, suggesting that the amplification contained both WT and *sdrM3** alleles. Additionally, 1.7a and 6c had mutations in a few other non-canonical genes, but these mutations did not affect resistance to DLX (Supplementary Dataset 2, Supplementary Fig. 8).

The *sdrM*, *lmrS*, and *sepA* genes showed a 3–10-fold increase in copy number in 1.7a, 4.2a, and 6c, while a control evolved isolate, 3a, that was predicted to not have the amplification, showed a similar copy number to the WT, as measured by genomic DNA qPCR and coverage data from the whole-genome sequencing (Fig. 3b, c, Supplementary Fig. 9). Further, 1.7a, 4.2a and 6c showed a 5–500-fold increase in gene expression of the three efflux pumps compared to the WT strain, as measured by quantitative reverse transcription PCR (RT-qPCR) (Fig. 3d). The dramatically higher expression compared to the copy number, especially in 1.7a and 4.2a, likely reflects altered regulation of the efflux pumps in the amplifications. All three isolates had ~5–100-fold higher DLX resistance and high DLX efflux compared to the parental WT strain (Fig. 3e, f).

To test whether the overexpression of the amplified efflux pumps individually, or in combination, can increase DLX resistance, we overexpressed *sepA*, *lmrS*, or the parental or evolved *sdrM* alleles under the control of the corresponding native promoters using the pKK30 plasmid[42]. In these strains, the expression of the efflux pumps increased ~10–100 fold compared to an empty vector control (Supplementary Fig. 10a). Overexpression of the WT *sdrM* allele led to a ~2-fold increase in DLX efflux activity and resistance, while the *sdrM1**, *sdrM2**, and *sdrM3** alleles showed a ~4–6-fold increase (Supplementary Fig. 10b, c). Further, overexpression of *sepA* and *lmrS* did not lead to a significant increase in DLX resistance, and overexpression of all 3 efflux pumps led to similar resistance as overexpression of the WT *sdrM* allele indicating that *sdrM* overexpression is likely the main contributor to DLX resistance caused by the genomic amplifications (Supplementary Fig. 10b).

### Amplification copy number increases upon antibiotic exposure and leads to cross-resistance

Amplification instability commonly causes changes in copy number, and consequently expression, which may depend on the selective strength of environmental conditions[12,14]. To test the stability of the amplification, we passaged 1.7a and 6c in two concentrations of DLX as well as antibiotic-free media for two days and determined the *sdrM* read coverage for each passage using whole-genome sequencing. The copy number of *sdrM* increased upon passaging in DLX, and decreased upon passaging without the antibiotic, indicating that the *sdrM* copy number, and likely its expression, in the genomic amplifications is dependent on the level of DLX exposure (Fig. 4a, b). Further, the frequency of the two mutations in the *sdrM3** allele (A268S coding sequence mutation and an upstream mutation) in 6c increased upon DLX passaging and decreased upon passaging in the antibiotic-free media (Fig. 4a). The isolates that had been passaged for two days in the higher concentration of DLX, and had higher amplification copy number, also showed a ~2-fold increase in DLX resistance for 6c, and a mild increase for 1.7a (Fig. 4c, d).

Given the presence of three efflux pumps in the amplification, and their resulting high expression, we tested the resistance of 1.7a grown with and without DLX, against a panel of different antibiotics. While we did not see increased resistance against most antibiotics, 1.7a grown in DLX showed cross-resistance against the aminoglycoside streptomycin (Fig. 4e). Further, overexpression of all three efflux pumps, but not

*sdrM* alone, resulted in a similar increase in streptomycin resistance (Fig. 4f), indicating that streptomycin resistance also required increased expression of *lmrS* and *sepA*.

### *sdrM* is required for selection of amplifications and increases evolvability of DLX resistance

The *sdrM*::Tn strain had no additional mutations compared to the WT, showed similar growth to the WT in M63 (Supplementary Fig. 11a, b) and had an MIC ~2-fold lower than the WT (Fig. 5a), indicating that SdrM contributes to the intrinsic DLX resistance level of the WT MRSA strain. To understand the role of *sdrM* in the selection of the genomic amplifications, we evolved three independent populations each of the transposon mutant strain *sdrM*::Tn and the WT (as a control), in increasing concentrations of DLX, similar to our original evolution. We verified that the transposon insertion was stable during the evolution in all three *sdrM*::Tn evolved populations (Supplementary Fig. 11c). During the evolution, the populations grew in DLX concentrations ~640–1000x the respective MICs, and terminal evolved populations for both the WT and *sdrM*::Tn showed high DLX MICs (Supplementary Fig. 11d). We saw genomic amplifications of the *sdrM* locus in 2 out of the three independently evolved WT populations. While the 3rd population also showed junctions associated with the amplification, the copy number increase was lower than our 1.3-fold threshold to denote the presence of an amplification. Each WT population had a novel amplification type not seen in the original evolution (Supplementary Dataset 4). However, none of the *sdrM*::Tn populations had these amplifications (Fig. 5b). Further, unlike the WT populations, all sequenced populations from the intermediate and terminal passages of the *sdrM*::Tn evolutions had mutations in both the DNA gyrase and topoisomerase IV enzymes, suggesting that the high DLX resistance was due to dual target mutations, as opposed to amplifications (Fig. 5b).

Amplifications are known to be unstable and commonly underlie antibiotic heteroresistance in populations[43,44], while resistance due to changes in DNA sequence is predicted to be more stable and homogenous. We, therefore, tested the population heterogeneity for antibiotic resistance in the evolved isolates 1.7a and 6c that have genomic amplifications but no canonical target mutations, as well as an isolate from the final passage of an evolved *sdrM*::Tn population that has mutations in *gyrA*, *gyrB*, and *parE* (Fig. 5b, Supplementary Dataset 2). We grew the isolates overnight in antibiotic-free media and then determined the fraction of the population that could grow in different DLX concentrations. We observed that while the *sdrM*::Tn evolved isolate (Tn_3b) showed ~100% DLX resistance at all concentrations tested, 1.7a and 6c showed population heterogeneity, where only ~10% or fewer cells were resistant to DLX (Fig. 5c).

The presence of mutations in both target enzymes in all evolved *sdrM*::Tn populations (Fig. 5b) suggested that in the absence of *sdrM*, mutations in both targets are essential for high DLX resistance. We, therefore, tested whether the presence of *sdrM* affected the evolvability of DLX resistance, similar to the experiment testing the evolved *sdrM* alleles (Fig. 2c). We evolved 12 independent populations each of the WT and *sdrM*::Tn in fixed concentrations of DLX that were 2.5x the respective MIC for each strain (0.55 μg/ml for WT and 0.32 μg/ml for *sdrM*::Tn), allowing for an additional day of growth on day 5 to aid in resistance evolution. While 5 of the 12 WT populations evolved DLX resistance in 6 days, only 1 of 12 *sdrM*::Tn populations evolved resistance in 8 days, indicating that the presence of *sdrM* increases the evolvability of DLX resistance (Fig. 5d).

### *sdrM* mutations and amplification are prevalent in clinical isolates evolved for DLX resistance

We tested two *S. aureus* clinical isolates for the incidence of *sdrM* mutations and amplification upon the evolution of DLX resistance. The two *S. aureus* clinical isolates were originally isolated from two cystic fibrosis patients. CF001 is a MRSA strain of clonal complex 8 (CC8)

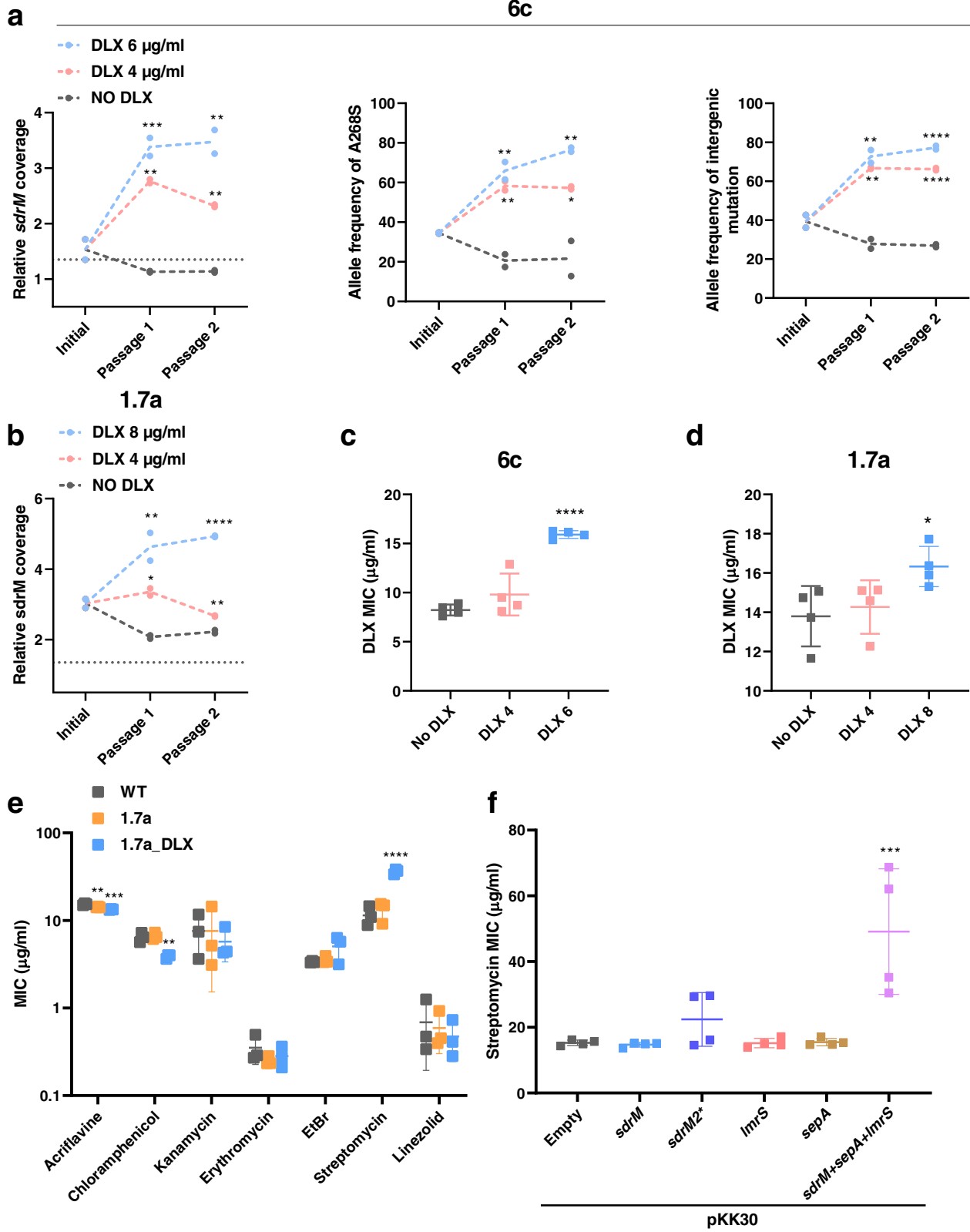

sequence type 8 (ST8) and CF106 is a CC1 ST188 MSSA strain (Supplementary Dataset 5). While CF001 had an DLX MIC ~2× lower than our WT strain JE2, the DLX MIC of CF106 was ~50× lower than JE2 (Fig. 6a). We evolved three independent populations each of both CF001 and CF106 in increasing DLX concentrations, and sequenced populations from intermediate passages (Fig. 6b). Two out of three populations of both CF001 and CF106 had the *sdrM2** allele,

while all CF001 and CF106 evolved populations had *sdrM* genomic amplifications.

## Discussion

Resistance against antibiotics with more than one cellular target likely requires multiple mutations in a cell and is thus predicted to be infrequent. However, the evolutionary paths leading to resistance

**Fig. 4 | Increased copy number of efflux pump amplification leads to cross-resistance. a** Copy number of *sdrM* and the allele frequencies of the two mutations that constitute the *sdrM3\** allele in 6c upon passaging in either no DLX media, or two concentrations of DLX. **b** Copy number of *sdrM* in 1.7a upon passaging in either no DLX media, or two concentrations of DLX. Data from two independent passaging experiments are shown for both. **c**, **d** DLX MICs in M63 of 6c and 1.7a populations after the second passage in either no DLX, 4 μg/ml DLX (DLX 4) or **c** 6 μg/ml DLX (DLX 6) for 6c and **d** 8 μg/ml DLX (DLX 8) for 1.7a. Data shown are the mean ± standard deviation of two biological replicates each from two independent passaging experiments. **e** MICs of WT, 1.7a, or 1.7a grown overnight in 2 μg/ml DLX (1.7a_DLX) for the indicated antibiotics in MH2. Data shown are the mean ± standard deviation of three biological replicates. **f** Streptomycin MICs of the pKK30-overexpression strains in MH2. Data shown are the mean ± standard deviation of

four biological replicates. Significance is shown for comparison to **a**–**d** the respective no DLX control, **e** WT and **f** the strain carrying the empty vector, as tested by a one-way ANOVA with the Holm−Sidak's test for multiple comparisons (*P < 0.05, **P < 0.01, ***P < 0.001, ****P < 0.0001). **a** For 'relative *sdrM* coverage' DLX 4 μg/ml: P for passage 1 = 0.0012, passage 2 = 0.0067, DLX 6 μg/ml: P for passage 1 = 0.001, passage 2 = 0.0018; for 'allele frequency of A268S' DLX 4 μg/ml: P for passage 1 = 0.0052, passage 2 = 0.0164, DLX 6 μg/ml: P for passage 1 = 0.0052, passage 2 = 0.0097; and for 'allele frequency of intergenic mutation' DLX 4 μg/ml: P for passage 1 = 0.0018, DLX 6 μg/ml: P for passage 1 = 0.0018; **b**, DLX 4 μg/ml: P for passage 1 = 0.0312, passage 2 = 0.0020, and DLX 8 μg/ml: P for passage 1 = 0.0092; **d** P for DLX 8 μg/ml = 0.048; **e**, acriflavine: P for 1.7a = 0.0025, 1.7a_DLX = 0.0002; chloramphenicol: P for 1.7a_DLX = 0.0035; **f** P for *sdrM+lmrS+sepA* = 0.0001. Source data are provided in the Source Data file.

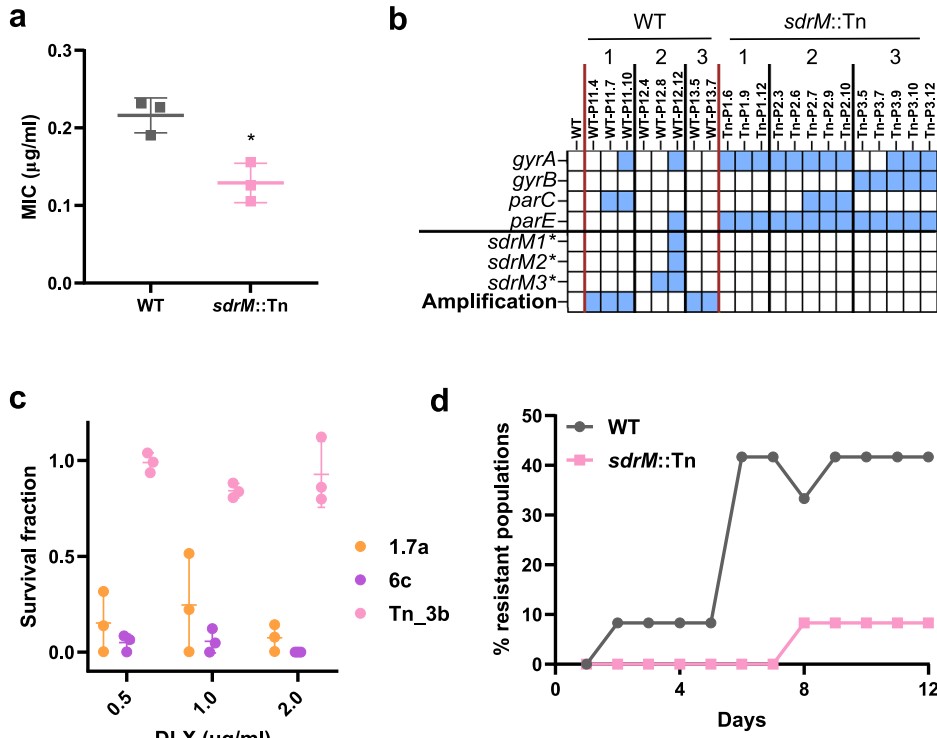

**Fig. 5 | The presence of *sdrM* facilitates the evolution of DLX resistance. a** DLX MICs of the WT and *sdrM*::Tn strains in M63. Data shown are the mean ± standard deviation of three biological replicates. Significance is shown for comparison to the WT as tested by an unpaired two-tailed *t* test (*P = 0.0112). **b** The presence of mutations in genes encoding DNA gyrase subunits (*gyrA, gyrB*) and DNA topoisomerase IV subunits (*parC, parE*), the three mutant alleles *sdrM1\**, *sdrM2\**, and *sdrM3\**, and a genomic amplification containing *sdrM*, are shown for populations from intermediate passages of three independently evolved populations each of

the WT and *sdrM*::Tn strains. **c** The survival fraction for the indicated isolates in multiple DLX concentrations compared to a no DLX control, measured as cfu/ml on the respective plates. Data shown are the mean ± standard deviation of three biological replicates. **d** Percentage of the 12 independent populations of the indicated strains that evolved DLX resistance over time in 2.5x the respective DLX MICs (0.55 μg/ml for the WT and 0.32 μg/ml for *sdrM*::Tn). Source data for (**a**, **c**, **d**) are provided in the Source Data file.

against such antibiotics are not well-characterized. In this study, we showed that the evolution of MRSA upon exposure to the dual-targeting 4th generation fluoroquinolone DLX led to resistance via both coding sequence and upstream mutations in *sdrM*, the gene encoding a poorly characterized major facilitator superfamily efflux pump, as well as gene amplifications of *sdrM*. Further, the presence of two additional efflux pumps adjacent to the *sdrM* locus, and consequently their hitchhiking in the genomic amplification, led to cross-resistance against the aminoglycoside streptomycin. In the absence of *sdrM*, strains required mutations in both canonical targets, DNA gyrase and topoisomerase IV, to attain DLX resistance, and therefore had reduced evolvability of DLX resistance. The *sdrM* mutations and amplifications thus provided a more accessible adaptive path to high DLX resistance. Our results suggest that antibiotics with multiple

targets may inadvertently lead to alternate adaptive trajectories that not only allow for rapid evolution of resistance, but also lead to additional unfavorable changes in the bacterial fitness landscape.

Efflux pumps are thought to provide rapid protection to cells upon antibiotic exposure, and efflux pump mutations are commonly associated with antibiotic resistance[45]. Several efflux pumps from multiple protein families are present in MRSA. The MFS family pumps include chromosomally encoded NorA, NorB, NorC, LmrS, MdeA, and SdrM, and plasmid-based overexpression of these pumps can confer resistance to fluoroquinolones, quaternary ammonium compounds, and dyes[46,47]. While mutations in *norA* have been associated with the evolution of antibiotic resistance, especially against fluoroquinolones[48], *sdrM* mutations have not been previously implicated. Our study thus shows that less well-characterized efflux

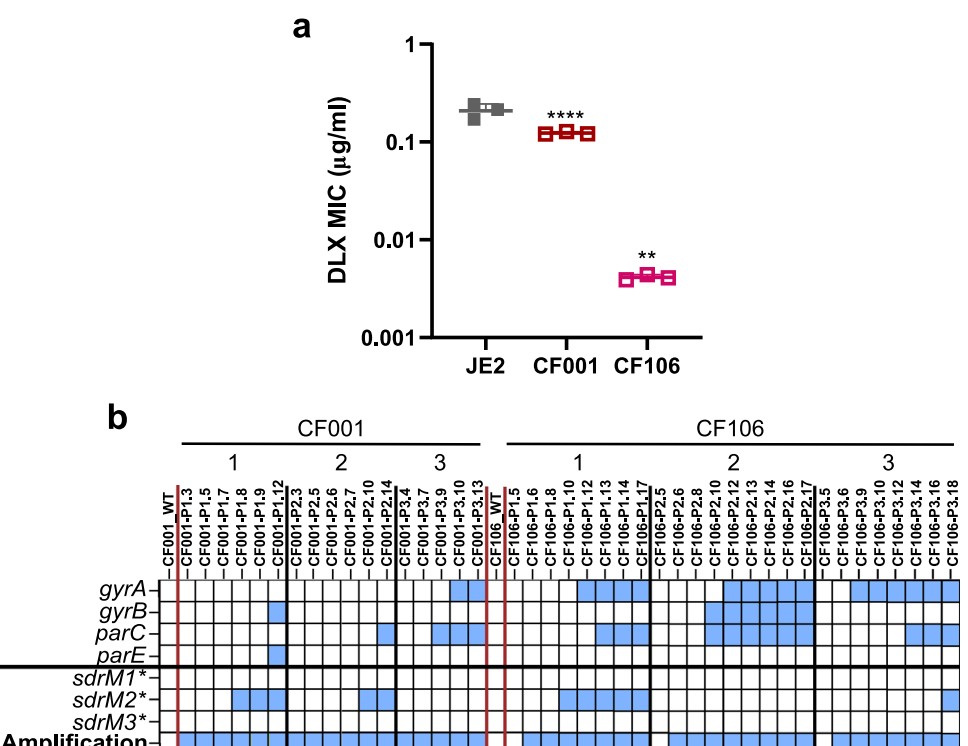

**Fig. 6 | Mutations and amplification of *sdrM* are pervasive in clinical isolates evolved for DLX resistance. a** DLX MICs of JE2 (WT) and the two clinical isolates CF001 and CF106. Data shown are the mean ± standard deviation of three biological replicates. Significance is shown for comparison to the WT as tested by a one-way ANOVA with the Holm−Sidak's test for multiple comparisons (**\*\*P = 0.0024, \*\*\*\*P < 0.0001*). **b** The presence of mutations in genes encoding DNA gyrase subunits (*gyrA*, *gyrB*) and DNA topoisomerase IV subunits (*parC*, *parE*), the three mutant alleles *sdrM1\**, *sdrM2\**, and *sdrM3\**, and a genomic amplification containing *sdrM*, are shown for populations from intermediate passages of three independently evolved populations of the clinical isolates CF001 and CF106. Source Data for **a** is provided in the Source Data file.

pumps may play a role in the acquisition of AMR, especially against newly developed antimicrobials. Further, it had been previously suggested that plasmid-based overexpression of *lmrS* and *sepA* in *E. coli* or a methicillin-sensitive *S. aureus* strain could lead to increased resistance against multiple antibiotics including linezolid, erythromycin, and kanamycin[39,40,49]. However, we did not observe cross-resistance against these antibiotics in our evolved isolates that overexpressed these efflux pumps, indicating that this resistance may be specific to *E. coli* or the strain of *S. aureus*.

Antibiotic resistance-associated efflux pump mutations typically increase the expression of the efflux pump, thereby reducing antibiotic concentrations within the cell. Such mutations are most often located either in the promoter region or in a repressor of the pump, although genomic amplifications of efflux pumps have also been observed[50,51]. In our evolutions, we did not identify any transcriptional repressor mutations that would increase *sdrM* expression, and only one population had a mutation upstream of *sdrM*, whereas *sdrM* amplifications were common in all ten independently evolved populations. This suggests that the substantial increase in *sdrM* expression required for high DLX resistance is largely inaccessible via promoter or repressor mutations. In the evolved isolates with amplifications, the expression levels of *sdrM* and the adjacent efflux pumps *lmrS* and *sepA* were much higher compared to the copy number of the coding genes (Fig. 3c, d) suggesting that the regulation of these genes is altered in the amplification. Interestingly, a cluster of tRNA-rRNA genes, which are typically extremely highly expressed, is located downstream of *sdrM*, *lmrS*, and *sepA*, but gets positioned upstream of the amplified copies of the efflux pump encoding genes. This raises the possibility that read-through transcription from the tRNA-rRNA genes may result in the highly increased efflux pump expression we observed in our evolved

isolates containing amplifications, similar to a previous observation in *Streptococcus pneumoniae*[51].

Coding sequence mutations in efflux pumps have also been shown to increase antibiotic resistance in *Pseudomonas aeruginosa* (*mexB*, *mexY*), *Klebsiella pneumoniae* (*kmrA*), and *Neisseria gonorrhoeae* (*mtrD*), likely due to increased affinity for the antibiotic[52–54]. Recent studies have suggested that increased expression of efflux pumps could also facilitate antibiotic resistance by increasing the mutation rate[55,56], allowing expression of resistance determinants upon plasmid acquisition[57], or promoting selection of resistance mutations due to increased fitness upon antibiotic exposure of strains both carrying the resistance mutations and overexpressing efflux pumps[58]. We also found that coding sequence mutations in the SdrM binding pocket increased DLX resistance and efflux, likely by altering the binding to DLX. The *sdrM3\** upstream mutation probably led to elevated levels of the *sdrM* efflux pump, which in turn led to a larger increase in DLX efflux and resistance (Fig. 2a, b). Further, the moderate increase in DLX resistance of the *sdrM* mutant alleles prompted increased evolvability of DLX resistance compared to the WT, which suggests that coding sequence efflux pump mutations might also be able to facilitate the evolution of additional mutations that lead to higher levels of resistance, possibly due to a synergistic increase in resistance, or elevated fitness, of combination mutants. Finally, we showed that amplification of *sdrM* provided a single-step adaptive route to DLX resistance, while at least two mutations, in the DNA gyrase and topoisomerase IV enzymes, were necessary for high DLX resistance in the absence of *sdrM*, and the presence of a functional *sdrM* gene, therefore, increased the evolvability of DLX resistance. While the eventual level of DLX resistance attained was similar between the WT and *sdrM*::Tn strains (Supplementary Fig. 11d), the presence of *sdrM* in the WT allowed for more rapid and frequent evolution of DLX

resistance (Fig. 5d). Since genomic amplifications are thought to be much more common than mutations in the genomic sequence[12,14], it is likely that the combination of an *sdrM* coding sequence mutation and amplification also arose more frequently than the combination of coding sequence mutations in the two target enzymes.

Multiple mutations contributed to DLX resistance in our evolved populations, of which the most important determinants are likely the *sdrM* evolved alleles, *sdrM* amplifications, and the canonical mutations in the DNA gyrase and topoisomerase subunits. Within these categories, the specific *sdrM*, *gyrA*, *gyrB*, *parC*, and *parE* alleles present, as well as the *sdrM* amplification copy number, likely affect the exact levels of resistance seen. Single mutations in either the gyrase or topoisomerase enzymes, as well as single copies of the evolved *sdrM* alleles lead to a 2–4-fold increase in the DLX MIC (Supplementary Fig. 1b, Fig. 2a), while mutations in both the DNA gyrase and topoisomerase IV enzymes likely lead to high DLX MICs, 40–250-fold higher than the WT, as seen in the evolved *sdrM*::Tn populations (Fig. 5b). Amplifications of *sdrM* similarly lead to high DLX resistance, resulting in DLX MICs 5–100-fold higher than the WT (Fig. 3e). We observed 3–9-fold higher DLX resistance in our *sdrM* overexpression strains (Supplementary Fig. 10b), which was lower than the resistance in the evolved isolates 1.7a, 4.2a, and 6c (Fig. 3e). This was likely due to the significantly higher *sdrM* expression seen in some of these evolved isolates (Fig. 3d, Supplementary Fig. 10a), as well as the increase in copy number of the *sdrM* amplification upon DLX exposure and subsequent increases in *sdrM* expression and DLX MICs (Fig. 4). The plasmid-based *sdrM* overexpression is unlikely to recapitulate the dynamic nature of the genomic amplifications, thus preventing a direct comparison of resistance levels between the evolved isolates and constructed allele-replacement and overexpression mutants.

There is significant diversity among *S. aureus* strains that are isolated in the clinic[59], but our experiments show that upon DLX exposure, selection for *sdrM* genomic amplifications, and at least the *sdrM2*\* allele, is common even in clinical isolates from different clonal complexes. DLX has only recently (in 2017) been approved for clinical use by the FDA[26] and therefore selection of DLX resistance in the clinic, and subsequent sequencing of resistant isolates, is unlikely to be common as yet. The few DLX-resistant *S. aureus* clinical isolates that were identified and sequenced in a previous study had mutations in both the DNA gyrase and topoisomerase IV enzymes[28]. However, these isolates were collected prior to DLX approval and clinical use, and the selective pressures and mutational paths that led to the acquisition of the target mutations are therefore unknown. Another recent study examining DLX-resistant clinical *S. aureus* isolates also identified mutations in the DNA gyrase and topoisomerase IV subunits, and indicated a role for efflux pumps in DLX resistance in at least one isolate[29]. Our analysis of more than 63,000 *S. aureus* genomes also showed that while A268S and Y363H mutations are rare, other mutations in SdrM, including in residues predicted to be in the binding pocket, are abundant. These mutations may potentially affect *S. aureus* resistance, and the evolution of resistance, to DLX and other antibiotics.

Genomic duplications and amplifications are estimated to be present in 10% of bacterial cells in growing cultures and due to their high-frequency, are thought to play an important role in the early adaptation to stress, including antibiotic resistance[12,14]. Our study demonstrates that genomic amplifications of efflux pumps can lead to antibiotic resistance and may be the first steps of evolutionary paths that allow for the rapid evolution of more stable resistance mutations, especially when multiple such mutations are required for high levels of resistance.

Genomic amplifications are typically unstable and frequently lost in the absence of selective pressure, leading to population heterogeneity, and antibiotic heteroresistance[43,44]. We observed similar amplification instability in our study, leading to population heterogeneity of DLX resistance. Further, treatment with high concentrations

of DLX led to increased amplification copy number, and consequently increased DLX resistance, as well as cross-resistance against streptomycin. Heteroresistance against fluoroquinolones was rarely seen in clinical isolates from four pathogens in a previous study and this was ascribed to resistance requiring target mutations[44]. However, our study shows that in MRSA, genomic amplifications of efflux pumps can lead to high fluoroquinolone resistance. The instability of genomic amplifications likely leads to their under-detection in laboratory and clinical studies, and their role in antibiotic resistance is thus probably underappreciated.

We quantified the copy number of the amplification using both qPCR on genomic DNA, as well as coverage depth in our whole-genome sequencing samples. However, given the amplification instability, and consequent heterogeneity even within an otherwise isogenic population of a single strain, long read sequencing with high coverage depth may be necessary to provide a more precise measurement of the copy number distribution of the amplification in a population.

Interestingly, 16 distinct genomic amplification types of *sdrM* were observed in our 13 independent WT populations, and the amplifications were dynamic within each population, further highlighting their instability. Additionally, we observed the same amplification arising and being selected for in multiple independent populations, suggesting that DNA breaks and amplifications may preferentially occur at certain genomic sites. The sequence of the amplification junctions consisted of either microhomology of at most 13 bp, or no homology, which is less than the 20–40 bp threshold required for RecA-mediated recombination[14]. This suggests that the initial duplication in our evolved populations likely occurred via non-homologous end-joining, or alternative end-joining, which have not been reported in *S. aureus* to date[60].

DNA breaks contribute to the formation of genomic amplifications, both in bacteria as well as eukaryotic systems[14,61]. DNA gyrase and topoisomerase IV enzymes relieve topological stress in the DNA, generating short-lived DNA double-stranded breaks, that are then re-ligated. Fluoroquinolones typically bind DNA gyrase or topoisomerase IV in a complex with DNA and may both promote DNA cleavage and strongly inhibit religation of the DNA ends, resulting in increased DNA double-stranded breaks[62,63]. Fluoroquinolone antibiotics may thus promote the formation of genomic amplifications, and in our study, we observed pervasive generation and selection of *sdrM*-containing genomic amplifications upon DLX exposure. A recent study also showed that exposure to the DNA gyrase poison albicidin which also blocks DNA religation[64] led to genomic amplifications of an inhibitor protein in *Salmonella* Typhimurium and *Escherichia coli*, and provided albicidin resistance, further indicating that inhibition of DNA topoisomerase enzymes may aid in the formation of genomic amplifications[65].

Multi-target antibiotics or drug combinations are thought to reduce the frequency of antibiotic resistance. However, our study indicates that if one of the targets of such treatments is the DNA gyrase or topoisomerase enzyme, it may instead select for gene amplifications that arise due to breaks in the DNA and lead to rapid evolution of resistance, and these amplifications may also lead to additional unanticipated fitness consequences, including cross-resistance. A better understanding of the genetic and environmental factors that affect genomic amplifications, as well as the adaptive trajectories that are accessible due to the selection of such amplifications, especially in clinical settings, is, therefore, necessary to inform new antimicrobial therapies.

## Methods
### Strains and growth conditions
All strains and plasmids used in this work are described in Supplementary Table 1. The clinical isolates were obtained from the Cystic

Fibrosis Foundation Isolate Core at Seattle Children's Hospital and were originally isolated from two different cystic fibrosis patients. Distribution of these isolates is covered by a Seattle Children's approved IRB protocol #14977.

All experiments in liquid media were carried out at 37 °C, shaking at 300 rpm, in modified M63 media (13.6 g/L KH$_2$PO$_4$, 2 g/L (NH$_4$)$_2$SO$_4$, 0.4 μM ferric citrate, 1 mM MgSO$_4$; pH adjusted to 7.0 with KOH) supplemented with 0.3% glucose, 0.1 μg/ml biotin, 2 μg/ml nicotinic acid, 1× Supplement EZ (Teknova) and 1× ACGU solution (Teknova), or in Mueller Hinton Broth 2 (MH2) (Millipore sigma) with or without 1× ACGU. For cloning and strain construction, strains were grown in LB liquid media (10 g/L bacto-tryptone, 5 g/L yeast extract, 10 g/L NaCl) or on LB plates (containing 15 g/L agar), supplemented with the appropriate antibiotic (10 μg/ml chloramphenicol, 50 μg/ml ampicillin, 10 μg/ml trimethoprim) or 0.4% para-chlorophenylalanine (PCPA) for counter-selection. MH2 plates were made with MH2-agar (Millipore sigma) and supplemented appropriately with 1× ACGU and DLX (either 0.5, 1, or 2 μg/ml).

### Laboratory evolution of DLX resistance
Ten independent populations of *S. aureus* JE2 cells were grown shaking at 300 rpm at 37 °C in modified M63 media, in the presence of increasing concentrations of DLX, with a daily 50–100-fold dilution into 14 ml snap-cap tubes containing 2 ml fresh media with DLX. The evolution started between 0.05–0.25 μg/ml DLX, and the concentration was increased 1.5–2-fold at each exposure. If the strains did not grow at the starting concentration, a lower DLX concentration was chosen to initiate the evolution. During the evolution, if a population did not show visible growth after one day, it was allowed an additional day of growth. If there was still no growth after two days, the evolution was reset to the previous passage, and a smaller DLX increment was chosen for the subsequent passage. Evolution was stopped either at 128 μg/ml or at any passage after 8 μg/ml if the populations didn't grow in the subsequent concentration in two attempts. Details of the populations are in Supplementary Dataset 1.

A similar setup was used to evolve the *sdrM*::Tn mutant, where three independent populations each of *sdrM*::Tn and WT were evolved in parallel, starting at a DLX concentration of 0.1 μg/ml, and the concentrations were increased 1.5–2-fold at each exposure. Similarly, for the evolution of the clinical isolates, three independent populations of CF001 and CF106 were evolved in parallel starting at a DLX concentration of 0.1 μg/ml for CF001 and 0.002 μg/ml for CF106. The concentrations were increased 1.5–2-fold at each exposure. Evolution was stopped as described above.

### Strain nomenclature
Each of the ten independently evolved populations from the initial evolution was labeled from P1-P10. Hence, P1 signifies the 1st independently evolved population. We introduce a dot and a second number which represents the passage number, for example, P1.7 indicates the 7th passage of the 1st evolved population. The letters appearing at the end of each number indicate that it was an isolate, for instance, 1.7a indicates an isolate extracted from population P1.7. The isolates from the final passage are represented as the population number and a letter, for example, 1a is an isolate from the final passage in population P1.

### Whole-genome sequencing of evolved populations and isolates
Genomic DNA was prepared from selected populations and isolates using the Qiagen DNeasy Blood and Tissue kit. Then, indexed single-end or paired-end libraries were prepared using the Illumina Nextera XT DNA Library Preparation kit and sequenced either on an Illumina MiSeq or Nextseq 500 sequencer. The data were analyzed as described before[66,67], where the Illumina adapters were removed using *cutadapt* v4.0[68], the reads were trimmed with *trimmomatic* v0.39[69], or both

steps done using *fastp* v0.23.2[67], and the mutations in the evolved populations and isolates were identified using *breseq* v0.37.1[70]. The mutations identified in the evolved populations in genes encoding the canonical targets or efflux pumps compared to the parental strain, as well as all mutations present in the populations from the terminal passage of each independently evolved population are listed in Supplementary Dataset 2. Mutations identified in a population at a frequency of at least 30% were considered as present.

### Analysis of clinical isolate genomes
The genomes of the clinical isolates CF001 and CF106 were assembled using *Unicycler v0.4.8*[71] on PATRIC[72] (now integrated into the Bacterial and Viral Bioinformatics Resource Center). The assembled contigs were then annotated using *Prokka v1.14.6*[73]. These annotated genomes were used as a reference to identify mutations in the respective evolved populations using *breseq*.

MLST of CF001 and CF106 was done using the PubMLST database online (pubmlst.org)[74], and the SCCmec typing was done using *Staphopia-sccmec*[75].

### Determination of genomic amplification copy number
The coverage depth of each base pair in the whole-genome sequencing data was determined using the BAM2COV command in *breseq* v0.37.1. The coverage of all base pairs was summed for the whole genome or each gene of interest and then was divided by the sequence length (number of base pairs). The fold change in coverage was then determined by dividing the corresponding number of reads per base pair of the gene with that of the whole genome. To identify *sdrM* amplifications, samples had to have a genomic junction which led to *sdrM* amplification (as determined by *breseq*) and meet a threshold for relative read coverage of *sdrM*. For isolates, the relative coverage of *sdrM* had to be at least 2× the mean WT value (from three biological replicates), while for populations, the relative *sdrM* coverage had to be at least 1.3× the mean WT coverage (to allow for population heterogeneity). For each amplification type, at least one instance was verified by PCR amplification of the novel junction followed by Sanger sequencing.

For the data shown in Supplementary Fig. 9, we calculated the normalized coverage of *sdrM*, *lmrS*, *sepA*, and *rpoC* as the sum of the reads that mapped to the gene divided by the gene length. The efflux pump normalized coverage was then divided by that of *rpoC*, and for each evolved isolate, normalized to the WT value to get the fold-change in coverage.

For the clinical isolates, as the genomes were assembled into contigs, we compared the coverage of *sdrM* to the contig that contains it. The coverage depth of each base pair in the contig was determined as above. The fold-change in *sdrM* coverage was determined by dividing the number of reads per base pair of *sdrM* by that of the contig. The criteria to identify an amplification were the same as above, except that the relative coverage of *sdrM* was compared to the value of the respective parental clinical isolate.

### Construction of allele-replacement mutants
Genomic DNA was extracted from evolved isolates, and the evolved alleles were amplified through PCR and cloned into pIMAY* digested with BamHI and EcoRI, using Gibson assembly[76]. The methylation profile of the plasmids was matched to *S. aureus* after electroporating and extracting the constructed plasmids from *E. coli* IM08B[77]. Plasmid extraction was done using Qiagen Plasmid MIDI kit and the plasmids were concentrated using Savant SpeedVac SPD1030. Allelic replacement was carried out in *S. aureus* JE2 as described[76]. Briefly, plasmids were electroporated into *S. aureus* JE2 cells and plated at 30 °C on LB Agar + chloramphenicol. Transformed colonies were grown shaking at 37 °C in LB media + chloramphenicol for two overnights, with a 1:1000 dilution into fresh medium after the first overnight. The following day

the cells were plated on LB Agar + chloramphenicol, and colonies were tested for integration by PCR with integration primers listed in Supplementary Table 2. For plasmid excision, colonies that showed integration were grown in LB media at 25 °C with shaking for two overnights, with a 1:1000-fold-dilution after the first overnight. Cells were then plated on LB plates containing 0.4% PCPA, and colonies were screened based on the size, with larger sizes assumed to indicate plasmid excision. 60–100 large colonies were streaked onto LB plates with and without chloramphenicol to identify chloramphenicol-sensitive clones. Presence of the mutant allele was confirmed by Sanger sequencing using the sequencing primers (see Supplementary Table 2).

## Analysis of allelic diversity of SdrM
Genome assemblies of 63,986 *S. aureus* isolates were downloaded from the NCBI Pathogen Detection database (https://www.ncbi.nlm.nih.gov/pathogens/) which was accessed on 19th January 2023. The FASTA sequences were concatenated, and a local BLAST database was created with the NCBI BLAST v2.13.0+ makeblastdb command[78]. The protein sequence of the JE2 SdrM sequence was used as a query for a tblastn (translated BLAST) search in the local database, and the output was parsed for all hits that had 100% coverage, a percent identity >85% and an E value < 10^−6. The amino acid frequencies for all positions were quantified from the tblastn BTOP output and are shown in Supplementary Dataset 3.

## Construction of overexpression strains
To construct the overexpression strains, WT genes, and *sdrM* mutant alleles were amplified by PCR and cloned into PCR amplified pKK30 vector using Gibson assembly. The Gibson assembled product was electroporated into *E. coli* DH5α λ-pir electrocompetent cells and recovered on LB Agar + trimethoprim plates. Plasmids were extracted from the *E. coli* strains and electroporated into *S. aureus* RN4220. Plasmids were extracted once more and electroporated into *S. aureus* JE2.

## Efflux assay
Cells were grown for 16 h in 2 ml of M63. 2 μl of these cells were added to 198 μl of M63 + 0.1 μg/ml DLX in a 96-well plate. The fluorescence of DLX (excitation and emission wavelengths of $\lambda_{exc}$ = 395 nm and $\lambda_{em}$ = 450 nm respectively), and the $OD_{600}$ were measured every 30 min for 19.5 h. Background readings were taken for cells in M63 with no DLX or 0.1 μg/ml DLX, and M63 alone with no DLX or 0.1 μg/ml DLX. Since the raw fluorescence was seen only in the presence of cells (Supplementary Fig. 4a), this likely indicates DLX accumulation inside cells, and can be used as a proxy for intracellular DLX. To enable comparison between samples, we normalized the fluorescence for cell density, and subtracted the autofluorescence of the media and cells. The normalized fluorescence (fluorescence/$OD_{600}$) was calculated as,

$$\frac{(Fluor.\,of\,cells\,in\,M63 + DLX) - (Fluor.\,of\,M63 + DLX)}{(OD_{600}\,of\,cells\,in\,M63 + DLX) - (OD_{600}\,of\,M63 + DLX)}$$
$$-\frac{(Fluor.\,of\,cells\,in\,M63) - (Fluor.\,of\,M63)}{(OD_{600}\,of\,cells\,in\,M63) - (OD_{600}\,of\,M63)}$$

Since the cell density is very low initially, and the cells are in lag phase, the normalized fluorescence at the early time-points shows high variability due to the low values of the denominator terms. We are therefore showing the data from the final 12.5 h in the figures.

## Model to determine the rate of DLX efflux
To determine the rates of DLX efflux, we considered a simplified mathematical model to only capture the increase in the DLX concentration inside the cells. Given the selection of *sdrM* mutations, and the effect of *sdrM* overexpression on DLX resistance, we assumed that

DLX efflux was mainly dependent on SdrM activity. Further since SdrM is an efflux pump, and unlikely to affect DLX metabolism or degradation, these processes should be similar across our tested strains, and we therefore did not include these in our model. We defined the following differential equations:

$$\frac{dC_{in}}{dt} = \rho_{in}C_{out} - \rho_{out}C_{in} \tag{1}$$

$$\frac{dC_{out}}{dt} = A - \rho_{in}C_{out} + \rho_{out}C_{in} \tag{2}$$

where, $C_{in}$ is the DLX concentration inside the cell at time $t$, $\rho_{in}$ is the rate of influx of DLX in to the cell, $C_{out}$ is the DLX concentration outside of the cell at time $t$, $\rho_{out}$ is the rate of efflux of DLX out of the cell and $A = C_{out}$ ($t$ = 0) is the initial DLX concentration we used for the experiment, 0.1 μg/ml (-0.226 μM). We assumed that DLX binding and unbinding occurs very fast, and do not take it into account in our equations for simplicity. Solving the two equations above we get:

$$\frac{d^2C_{in}}{dt^2} + (\rho_{in} + \rho_{out})\frac{dC_{in}}{dt} - \rho_{in}A = 0 \tag{3}$$

Integrating this second order linear ordinary differential equation with $C_{in}$ ($t$ = 0) = 0 we get,

$$C_{in} = \frac{1}{(\rho_{in} + \rho_{out})}(B(e^{-(\rho_{in} + \rho_{out})t} - 1) + \rho_{in}\,A\,t) \tag{4}$$

where B is an arbitrary constant. As the normalized fluorescence we determined with the efflux assay was a proxy to $C_{in}$, we performed a least-squared fit of $C_{in}$ with the normalized fluorescence vs time for *sdrM*::Tn, to determine B and $\rho_{in}$, assuming that the rate of efflux will be equal to the rate of influx, $\rho_{in} = \rho_{out}$ for the transposon mutant since SdrM is absent and the transport of DLX into and out of the cell will be passive. The best-fit values for B and $\rho_{in}$ were assumed to be the same for the other strains. Substituting these values, we performed least-squared fit for WT, *sdrM1**, *sdrM2**, and *sdrM3** to determine the best-fit values for the efflux rates $\rho_{out}$ for each strain. As we were fitting $C_{in}$ to the normalized fluorescence, the units of the rates of influx and efflux were arbitrary units/time (hours).

## Measurement of MICs
DLX was serially diluted 2-fold in a Corning 96-well flat clear bottom plate, to obtain eight concentrations. Cells grown overnight in M63 media or Mueller Hinton Broth 2 (MH2) were transferred at a final dilution of 1:5000 to the 96-well plate containing the serially diluted antibiotic and grown at 37 °C with shaking for 24 h. After growth, the $OD_{600}$ of all wells was measured using a Biotek Synergy H1 microplate reader. The $OD_{600}$ measurements for all DLX concentrations were plotted against the DLX concentration values, and fitted to a modified Gompertz function to determine the exact MIC values[79]. To determine the multidrug MICs, WT was grown in MH2 broth supplemented with 1× ACGU, and 1.7a was grown in MH2 supplemented with 1× ACGU with or without 2 μg/ml DLX overnight. The following day, all three overnights were washed twice in 1× PBS and the MICs for all antibiotics were determined. MH2 medium was supplemented with 1× ACGU for experiments with 1.7a, as *pyrC*, a gene involved in pyrimidine synthesis, had a frameshift mutation in 1.7a, resulting in poor growth without 1× ACGU supplementation.

## Quantitative PCR (qPCR)
Genomic DNA was extracted as described above. RNA was extracted from cells grown overnight in M63 using the Total RNA Purification Plus Kit (Norgen Biotek Corp) and DNA was removed using the TURBO

DNA-free™ Kit (Invitrogen). cDNA was synthesized using random primers and Superscript III Reverse Transcriptase (Thermo Fisher Scientific). Genomic DNA or cDNA samples were mixed with primers (see Supplementary Table 2) and Applied Biosystems Power SYBR Green PCR Master Mix (Thermo Scientific) in a Microamp EnduraPlate Optical 384 Well Clear Reaction Plate (Thermo Fisher Scientific) and the qPCR or RT-qPCR, respectively, was run on a QuantStudio 5 real-time PCR machine (Thermo Fisher Scientific). The *rpoC* gene was used as the housekeeping control for both the qPCR and RT-qPCR[80].

### Measurement of evolvability

Twelve independent colonies for each strain were inoculated in a 96-well plate containing 200 μl of fresh M63 at 37 °C with shaking for 24 h. The next day, 10 μl of each was inoculated in 190 μl of fresh M63 containing 2.5 times the respective DLX MICs (*sdrM*::Tn at 0.32 μg/ml, WT at 0.55 μg/ml, *sdrM1** and *sdrM2** at 1 μg/ml, and *sdrM3** at 1.75 μg/ml) and grown for 23 h at 37 °C with shaking. After 23 h, $OD_{600}$ was measured and 10 μl of each well was transferred to a 96-well plate containing 190 μl of fresh M63 and DLX, and this was repeated for 5–14 days. For the evolution of the WT and *sdrM*::Tn strains (data shown in Fig. 5e), the populations were grown for an additional 24 h after Day 5 without transfer, to potentially allow for additional growth and evolution, to enable comparisons between the two strains. Populations that had an $OD_{600} \geq 0.400$ and maintained this growth consistently until the end of the experiment were classified as resistant. The experiment was stopped if there was no growth for three consecutive days in any of the wells.

### Determination of amplification stability

On day 1, strains were inoculated into fresh M63 media and grown for 24 h at 37 °C with shaking without DLX. On day 2, cells were diluted 1000-fold into tubes containing fresh media with no antibiotic or DLX at the appropriate concentrations. The same process was repeated for day 3, and for each strain there were two replicates. Genomic DNA was extracted each day, and whole-genome sequencing was performed. The *sdrM* copy number was determined as described above.

### Determination of population heterogeneity of DLX resistance

Cells were grown overnight in MH2 broth supplemented with 1× ACGU, and the following day serial dilutions of the cultures were spotted on to MH2-agar plates supplemented with 1× ACGU, without DLX, or containing DLX at 2 μg/ml, 1 μg/ml, or 0.5 μg/ml and grown in a 37 °C incubator overnight. The following day, colony-forming units were determined.

### Determination of growth parameters of WT and *sdrM*::Tn

Cells were grown overnight in M63, diluted 1:2500 in fresh M63, and 200 μl was transferred into a 96-well plate. Plates were incubated in a Biotek Synergy H1 microplate reader at 37 °C with continuous shaking at 807 cycles per minute, and the $OD_{600}$ of the plate was measured every 30 min for 24 h. To determine the cell density, serial dilutions of overnight cultures were spotted on LB agar plates and grown in a 37 °C incubator overnight. The following day, colony-forming units were determined.

### Statistical analysis

Statistical tests and analysis were performed in GraphPad Prism v8.4.3.

### Reporting summary

Further information on research design is available in the Nature Portfolio Reporting Summary linked to this article.

## Data availability

Genome assemblies of *S. aureus* isolates to analyze allelic variability of SdrM were downloaded from the NCBI Pathogen Detection database (https://www.ncbi.nlm.nih.gov/pathogens/). The whole-genome sequencing data from this study have been deposited at the NCBI Short Read Archive (SRA), associated with the BioProject PRJNA904786. Source data are provided in this paper.

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

## Acknowledgements
We would like to acknowledge the Center for Cancer Research Genomics Core for whole-genome sequencing library preparation and sequencing, and the Bose lab (University of Kansas Medical Center) for providing the pKK30 plasmid. We thank Tiffany Zarrella for sequencing and assembling the genomes of the clinical isolates, as well as for assistance with the SCCmec typing. We thank Susan Gottesman, Gigi Storz, John Dekker, and members of the Khare lab for comments on the manuscript, and members of the Gottesman, Ramamurthi, and Khare labs for discussion and suggestions throughout this work. This study utilized the computational resources of the NIH High-Performance Computing Biowulf cluster (http://hpc.nih.gov). This work was supported by the Intramural Research Program of the NIH, National Cancer Institute, and Center for Cancer Research.

## Author contributions
G.S. performed the original evolution experiments, K.P.T.S. performed all other experiments, K.P.T.S. and A.K. analyzed data and wrote the manuscript, A.K. supervised the research.

## Funding

## Competing interests
The authors declare no competing interests.
