## [Peer review file · Nature Communications]

REVIEWER COMMENTS

Reviewer #1 (Remarks to the Author):

In this manuscript, the authors investigate resistance evolution of MRSA to the novel dual-target antibiotic delafloxacin. They show that resistance evolves repeatedly and to high levels due to de novo mutations. They describe the adaptive changes leading to increased resistance, namely known fluoroquinolone resistance mutations in the target genes as well as a so far unknown mechanism via sequence mutations and increased expression through gene amplifications of the efflux pump *sdrM*. They characterize the resistance increase caused by the respective mutations and amplifications of *sdrM* and show differential evolution depending on the *sdrM* allele.

Dual-target antibiotics have been proposed to reduce resistance evolution and could thereby help mitigate the antibiotic resistance crisis. It is therefore relevant to investigate the evolution of resistance to such drugs as done in the present work for one specific example. The authors describe a resistance mechanism to DLX via an understudied efflux pump. More generally, however, resistance via efflux pumps is very common and has been described for a variety of antibiotics and organisms as stated by the authors themselves. The main findings are not particularly provocative and well supported by multiple, complementary experiments.

Specific comments

1. The evolution experiment was performed using a simple protocol of transfers with a constant increase of antibiotic concentration. This setup does not control variables such as population size, doubling time, number of generations, and cannot resolve more fine-grained resistance increases. While sufficient to draw the proposed conclusions, in particular the effect of variable population size during the experiment on comparability of different populations and strains such as *sdrM*::Tn and wild type should be discussed.
2. The observed mutations and amplifications all increase resistance to DLX to a variable degree. It would be interesting to relate the observed resistance for each specific population or isolate from the evolution experiments to the effects of mutations and amplifications measured in the allele replacement and overexpression experiments. How much of the observed resistance is explained by the combination of individual mutations or amplifications?
3. The claim of increased evolvability for different *sdrM* alleles seems exaggerated; this result should be more clearly specified.

- a. The methodology used to determine the percentage of resistant populations (Figure 2C,D and 5D) should be clarified. Is a certain OD600 threshold used to distinguish resistant populations and no growth?
- b. As in Figure 5D, concentrations normalized to the respective initial MIC of the strain should have been used in Figure 2C,D as well. Otherwise, differences in population size due to variable initial resistance make the comparison of strains impossible. Populations benefit from an initially higher MIC that allows higher population sizes and survival long enough for other mutations to appear.
- c. Overall resistance levels in the second evolution experiment comparing wild type and sdrM::Tn do not appear to be different (Supp Table 1). MIC measurements of the evolved populations would be a helpful comparison here.

Minor comments

1. Figure 1: this overview of the experimental populations would benefit from showing a measure of final resistance as in Supplementary Figure 1A or the final concentration used for these populations.
2. Laboratory evolution of DLX resistance: The method description should be clarified
 - Which growth medium was used?
 - Why were starting concentrations and dilutions during passages variable?
 - The concentrations in Supplementary Table 1 do not entirely match the description of consistent two-fold increase for the initial experiment.
3. Whole genome sequencing analysis (methods): How frequent does a mutation or a mutant have to be to be detected or considered in the population? It is not clear if additional mutations were found in the WGS data of evolved populations.
4. Efflux assay (methods): Since this is not a standard method, the reasoning behind the fluorescence based efflux assay should be elaborated to clarify the following:
 - Why is the normalized fluorescence a good proxy for the intracellular DLX concentration?
 - Could DLX degradation occur inside the cells?
 - What is happening in the first 10 h of the assay that are not shown in the figures?

5. Figure 4E would be better visualized with the MIC on a log scale so that a visual comparison is possible for all antibiotics.

Reviewer #2 (Remarks to the Author):

The authors perform classical adaptive evolution experiments in presence of a last-generation quinolone that equally inhibit both quinolone targets and find that, besides mutations in target-encoding genes, amplifications (and mutations) in a new efflux pump are involved as well. The finding that this specific efflux pump is involved in resistance is novel, but the concept that efflux pumps are relevant for multi-target antibiotics is not. Actually, all quinolones, although not equally active are dual-target antibiotics and efflux pumps are involved in resistance. Further, they are involved in resistance to antibiotic combinations, each of the antibiotics presenting a different target. Taking these previous findings, the paper should be more appropriate for a specialized journal.

Specific issues:

It is not fully clear how amplification was quantified. From Supplemental Figure 5, it seems that amplification is not higher than 2-3-fold. However, in Figure 3 numbers are higher. Please clarify.

Also, in Figure 3, Copy number and level of expression do not always correlate: see for instance *lmrS* in 1.7a and 4.2a and the explanation of the high increase in expression with not so high increase in amplification is weak. Please discuss in more detail.

I guess that the authors have measured MIC by double dilution. If this is true, values of 2-fold change must be taken with caution, because this is the error of the method. Further MIC means Minimal Inhibitory Concentration. This means that, if the experiment is repeated, the MIC is not the average. It is the minimal concentration that is obtained. If several values are high and one is low, it is acceptable to discard this one, but for some values are high and some are low, an average cannot be used.

For small changes in MIC as here reported, it is however acceptable to make growth curves in presence of different concentrations of antibiotics, use intermediate concentration or Epsilon tests.

Reviewer #3 (Remarks to the Author):

The paper by Silva et al represents an interesting description of a novel mechanism of resistance to a 4th generation Fluoroquinolone Delafloxacin. The study appears to have been well executed, robust and with appropriate conclusions. The findings provide potentially important insights relevant to the development of antimicrobial resistance via conserved efflux pumps. I have some suggestions to improve the manuscript and to understand the broader clinical relevance of the discovery.

Major:

It would be extremely informative to examine the allelic diversity of the SdrM gene across the *S. aureus* species diversity to examine amino acid variation that may impact on function. For example, are the sdrM mutations identified via experimental evolution represented among clinical isolates? Are other mutations identified that may affect the binding pocket of the efflux pump? While I appreciate that the drug has not been in clinical use for long, it is feasible that resistant strains may have evolved already, and/or some strains may have sdrM mutations that may make them more susceptible to developing resistance.

The experimental evolution experiments described were carried out with the USA300 strain, it would be helpful to know if the described phenomenon relating to sdrM mutations and amplifications leading to resistance also applies to other strains/lineages of *S. aureus*.

Minor:

Is the sdrM-Tn insertion stable? Has the genome sequence of the sdrM-Tn mutant been determined to account for possible spurious mutations that could impact function?

Fig 1 legend requires clarification of which isolates represent intermediate v terminal passages. Also indicate that the term 'mutations can refer to a SNP or an amplification.

l.88 '3 individual isolates'- presumably of 3 sampled ie 100%?

Please make sure throughout the manuscript that the terms isolate, strain and population are used appropriately eg l.17- 'we observed that strains'...should this be isolates?

We are grateful to the reviewers for their valuable suggestions and comments that we believe have significantly enhanced our manuscript. We have responded to all of their concerns, performing several additional experiments to better define the effect of the evolved *sdrM* alleles on the evolvability of delafloxacin resistance, determine the growth phenotype and stability of the *sdrM::Tn* mutation, and test the generality of adaptive trajectories to DLX resistance involving *sdrM* mutations and amplification in clinical isolates, as well as analysis of 60,000+ *S. aureus* genomes to identify the sequence variation seen in SdrM. Additionally, we have clarified several methodological details, and added information to figures and tables. The major changes are:

- 1) A new evolution experiment for the WT and allele-replacement strains carrying the evolved *sdrM* alleles, at 2.5x their respective MICs, to test their evolvability of DLX resistance. This is shown in the new Figure 2C.
- 2) Evolution of DLX resistance in two clinical isolates (one MRSA, and one MSSA), and analysis of the observed mutations. This is shown in the new Figure 6.
- 3) Analysis of the allelic diversity of SdrM in a panel of ~64,000 *S. aureus* genomes. This is shown in the new Supplementary Table 3.
- 4) Testing the growth parameters of the *sdrM::Tn* mutant, stability of the transposon insertion, and DLX resistance of the *sdrM::Tn* evolved populations. This is shown in the new Supplementary Figure 11.
- 5) Calculation of the *sdrM* coverage from whole-genome sequencing data via normalization to the housekeeping gene *rpoC*, instead of the entire genome, to compare coverage values from WGS vs qPCR. This is shown in the new Supplementary Figure 9.

Our point-by-point responses to the comments are denoted below in bold type.

Reviewer #1 (Remarks to the Author):

In this manuscript, the authors investigate resistance evolution of MRSA to the novel dual-target antibiotic delafloxacin. They show that resistance evolves repeatedly and to high levels due to de novo mutations. They describe the adaptive changes leading to increased resistance, namely known fluoroquinolone resistance mutations in the target genes as well as a so far unknown mechanism via sequence mutations and increased expression through gene amplifications of the efflux pump *sdrM*. They characterize the resistance increase caused by the respective mutations and amplifications of *sdrM* and show differential evolution depending on the *sdrM* allele.

Dual-target antibiotics have been proposed to reduce resistance evolution and could thereby help mitigate the antibiotic resistance crisis. It is therefore relevant to investigate the evolution of resistance to such drugs as done in the present work for one specific example. The authors describe a resistance mechanism to DLX via an understudied efflux pump. More generally, however, resistance via efflux pumps is very common and has been described for a variety of antibiotics and organisms as stated by the authors themselves. The main findings are not particularly provocative and well supported by multiple, complementary experiments.

Specific comments

1. The evolution experiment was performed using a simple protocol of transfers with a constant increase of antibiotic concentration. This setup does not control variables such as population size, doubling time, number of generations, and cannot resolve more fine-grained resistance increases. While sufficient to draw the proposed conclusions, in particular the effect of variable population size during the experiment on comparability of different populations and strains such as *sdrM::Tn* and wild type should be discussed.

We agree with the reviewer that our evolution experiment did not control for population size or number of generations. The goal of the study was to identify potential adaptive trajectories that could lead to resistance against a dual-targeting antibiotic, first in the WT strain, and subsequently in an *sdrM::Tn* mutant, and as the reviewer states, our experiments were sufficient to draw the stated conclusions. We appreciate the reviewer's concern that differences in population size could affect the conclusions drawn in comparing the rates of evolution. For the comparison between WT and *sdrM::Tn* (data shown in Figure 5D), we started the initial populations from overnight cultures, and performed 1:20 dilutions thereafter at each passage – thus the initial population size could be different, and potentially affect the evolution rates. We therefore measured the growth of the two strains, as well as the cell densities of overnight cultures, and do not see significant differences between the two strains, suggesting that population size did not play a significant effect in the results. We now show and describe this in the text:

Lines 244-247: “The *sdrM::Tn* strain had no additional mutations compared to the WT, showed similar growth to the WT in M63 (Supplementary Figure 11A, 11B) and had an MIC ~2-fold lower than the WT (Figure 5A), indicating that SdrM contributes to the intrinsic DLX resistance level of the WT MRSA strain.”

2. The observed mutations and amplifications all increase resistance to DLX to a variable degree. It would be interesting to relate the observed resistance for each specific population or isolate from the evolution experiments to the effects of mutations and amplifications measured in the allele replacement and overexpression experiments. How much of the observed resistance is explained by the combination of individual mutations or amplifications?

We appreciate the reviewer's suggestion that given the distinct mutations contributing to DLX resistance, it would be instructive to analyze the contributions of these mutations to the resistance seen in the 10 evolved populations. The several distinct mutations seen in the canonical targets, as well as the different combinations of the canonical mutations, *sdrM* evolved alleles, and *sdrM* genomic amplifications, likely affect the resistance in each population. We have now discussed this in the manuscript:

Lines 366-384: “Multiple mutations contributed to DLX resistance in our evolved populations, of which the most important determinants are likely the *sdrM* evolved alleles, *sdrM* amplifications, and the canonical mutations in the DNA gyrase and topoisomerase subunits. Within these categories, the specific *sdrM*, *gyrA*, *gyrB*, *parC*, and *parE* alleles present, as well as the *sdrM* amplification copy number, likely affect the

exact levels of resistance seen. Single mutations in either the gyrase or topoisomerase enzymes, as well as single copies of the evolved *sdrM* alleles lead to a 2-4 fold-increase in the DLX MIC (Supplementary Figure 1B, Figure 2A), while mutations in both the DNA gyrase and topoisomerase IV enzymes likely lead to high DLX MICs, 40-250x higher than the WT, depending on the number and combination of mutations present, as seen in the evolved *sdrM::Tn* populations (Figure 5B). Amplifications of *sdrM* similarly lead to high DLX resistance, resulting in DLX MICs 5-100 fold higher than the WT (Figure 3E). We observed 3-9 fold higher DLX resistance in our *sdrM* overexpression strains (Supplementary Figure 10B), which was lower than the resistance in the evolved isolates 1.7a, 4.2a, and 6c (Figure 3E). This was likely due to the significantly higher *sdrM* expression seen in some of these evolved isolates (Figure 3D, Supplementary Figure 10A), as well as the increase in copy number of the *sdrM* amplification upon DLX exposure and consequent increases in *sdrM* expression and DLX MICs (Figure 4). The plasmid-based *sdrM* overexpression is unlikely to recapitulate the dynamic nature of the genomic amplifications, thus preventing a direct comparison of resistance between the evolved isolates and constructed allele-replacement and overexpression mutants.”

3. The claim of increased evolvability for different *sdrM* alleles seems exaggerated; this result should be more clearly specified.

a. The methodology used to determine the percentage of resistant populations (Figure 2C,D and 5D) should be clarified. Is a certain OD₆₀₀ threshold used to distinguish resistant populations and no growth?

We used a threshold of OD₆₀₀ = 0.400 to classify populations as resistant (growth) or sensitive (no growth). Further, to ensure that we were not seeing sporadic growth, we set an additional criterion that once it started growing, a population should consistently show growth until the end of the experiment to be classified as resistant. These details have now been added to the methods section.

Line 662-664: “Populations that had a OD₆₀₀ ≥ 0.400 and maintained this growth consistently until the end of the experiment were classified as resistant.”

b. As in Figure 5D, concentrations normalized to the respective initial MIC of the strain should have been used in Figure 2C,D as well. Otherwise, differences in population size due to variable initial resistance make the comparison of strains impossible. Populations benefit from an initially higher MIC that allows higher population sizes and survival long enough for other mutations to appear.

The reviewer raises an important point here that since we had performed our evolvability experiments for the WT and evolved *sdrM* alleles at the same delafloxacin concentrations, the increased evolvability we saw for the *sdrM strains may just be a function of their higher MICs.**

To address this concern, we performed a new evolvability assay by evolving WT and the *sdrM strains in delafloxacin concentrations 2.5x their corresponding MICs. We observed**

that even when evolved under MIC-normalized delafloxacin concentrations, populations from all evolved *sdrM* allele-replacement strains showed a higher frequency of resistance evolution, and thus, higher evolvability, compared to the WT, similar to the results we had seen previously. We have replaced our previous data with the results from this new experiment (Fig 2C), and changed the text accordingly:

Lines 156-161: “To test if *sdrM* mutations can affect the evolvability of resistance in the presence of DLX, we evolved 12 independent populations each of WT, and strains carrying the individual *sdrM* evolved alleles with daily passages in fixed concentrations of delafloxacin, that were 2.5x times the respective DLX MICs. While only 1 WT population evolved resistance, 3-4 populations each of *sdrM1**, *sdrM2**, and *sdrM3** evolved resistance during the experiment (Figure 2C).”

c. Overall resistance levels in the second evolution experiment comparing wild type and *sdrM::Tn* do not appear to be different (Supp Table 1). MIC measurements of the evolved populations would be a helpful comparison here.

The overall resistance levels reached at the end of the second evolution experiment (comparing the WT and *sdrM::Tn* mutant) were similar. We measured the MICs of the 3 final WT and *sdrM::Tn* evolved populations, and have added that data to the manuscript (Supplementary Figure 11D), and changed the text appropriately:

Lines 251-253: “During the evolution, the populations grew in DLX concentrations ~640-1000 times the respective MICs, and terminal evolved populations for both the WT and *sdrM::Tn* showed high DLX MICs (Supplementary Figure 11D)”

Minor comments

1. Figure 1: this overview of the experimental populations would benefit from showing a measure of final resistance as in Supplementary Figure 1A or the final concentration used for these populations.

We have added the final concentrations used in each population at the bottom of the overview (Figure 1).

2. Laboratory evolution of DLX resistance: The method description should be clarified

- Which growth medium was used?
- Why were starting concentrations and dilutions during passages variable?
- The concentrations in Supplementary Table 1 do not entirely match the description of consistent two-fold increase for the initial experiment.

We have added details of the experimental evolution to the methods section to better describe the growth medium, the variability in the starting concentrations/dilutions, and the DLX concentrations used.

Lines 472-481: “Ten independent populations of *S. aureus* JE2 cells were grown shaking at 300 rpm at 37°C in modified M63 media, in the presence of increasing concentrations of DLX, with a daily 50-100-fold dilution into 14 ml snap-cap tubes containing 2 ml fresh media with DLX. The evolution was started between 0.05-0.25 µg/ml DLX, and the concentration was increased 1.5-2 fold at each exposure. If the strains did not grow at the starting concentration, a lower DLX concentration was chosen to initiate the evolution. During the evolution, if a population did not show visible growth after one day, it was allowed an additional day of growth. If there was still no growth after two days, the evolution was reset to the previous passage and a smaller DLX increment was chosen for the subsequent passage. Evolution was stopped either at 128 µg/ml DLX or any time after 8 µg/ml if the populations didn’t grow in the next concentration in two attempts.”

3. Whole genome sequencing analysis (methods): How frequent does a mutation or a mutant have to be to be detected or considered in the population? It is not clear if additional mutations were found in the WGS data of evolved populations.

We did observe additional mutations in the evolved populations. We have now shown the mutations present in all the terminal evolved populations in Supplementary Table 2. To be considered as being present in a population, we set a threshold frequency of 30%. We have added this to the methods.

Line 510-511. “Mutations identified in a population at a frequency of at least 30% were considered as present.”

4. Efflux assay (methods): Since this is not a standard method, the reasoning behind the fluorescence based efflux assay should be elaborated to clarify the following:

- Why is the normalized fluorescence a good proxy for the intracellular DLX concentration?

Fluoroquinolones commonly have intrinsic fluorescence, and DLX has been previously reported to be fluorescent as well. We measured the raw fluorescence of a sub-MIC concentration (0.1 µg/ml) of DLX vs time in just media, or media containing WT cells, and observed intrinsic DLX fluorescence only when the cells were present (new Supplementary Fig 4A), indicating that accumulation of DLX inside cells leads to increased DLX fluorescence. We determined the fluorescence per unit OD₆₀₀ to normalize for the number of cells present in the sample and subtracted the background fluorescence of the cells and media from this. We defined this as the normalized fluorescence, which we thus believe is a good proxy for the intracellular DLX concentration. We have now added this information to the methods:

Line 586-591: “Background readings were taken for cells in M63 with no DLX or 0.1 µg/ml DLX, and M63 alone with no DLX or 0.1 µg/ml DLX. Since the raw fluorescence was seen only in the presence of cells (Supplementary Figure 4A), this likely indicates DLX accumulation inside cells, and can be used as a proxy for intracellular DLX. To enable

comparison between samples, we normalized the fluorescence for cell density, and subtracted the autofluorescence of the media and cells.”

- Could DLX degradation occur inside the cells?

It is unknown whether DLX degradation can occur inside cells, and if yes, the degradation rates are unknown. However, given that the difference in the strains we compare is primarily in SdrM, an efflux pump, it is likely that the rate of DLX degradation inside the cell, if any, would be comparable in all strains. We have mentioned this in the methods section.

Line 603-605: “Further since SdrM is an efflux pump, and unlikely to affect DLX metabolism or degradation, these processes should be similar across strains, and we therefore did not include these in our model.”

- What is happening in the first 10 h of the assay that are not shown in the figures?

At the start of the assay, the cell density is very low, and the cells are in lag phase, resulting in very low OD₆₀₀ values for the first several hours. For the normalized fluorescence, which has the differences in the optical densities in the denominator, the denominator is thus very small at these initial time-points, resulting in high variability in this value. This can be seen in the figure below where we are showing the mean values of the normalized fluorescence for the WT. We are therefore showing only the later time-points which have more reproducible values.

We have now mentioned this in the methods section:

Lines 595-597 “Since the cell density is very low initially, and the cells are in lag phase, the normalized fluorescence at the early time-points shows high variability due to the low values of the denominator terms. We are therefore showing the data from the final 12 hours in the figures.”

5. Figure 4E would be better visualized with the MIC on a log scale so that a visual comparison is possible for all antibiotics.

We thank the reviewer for this suggestion and have changed the figure accordingly (revised Figure 4E).

Reviewer #2 (Remarks to the Author):

The authors perform classical adaptive evolution experiments in presence of a last-generation quinolone that equally inhibit both quinolone targets and find that, besides mutations in target-encoding genes, amplifications (and mutations) in a new efflux pump are involved as well. The finding that this specific efflux pump is involved in resistance is novel, but the concept that efflux pumps are relevant for multi-target antibiotics is not. Actually, all quinolones, although not equally active are dual-target antibiotics and efflux pumps are involved in resistance. Further, they are involved in resistance to antibiotic combinations, each of the antibiotics presenting a different target. Taking these previous findings, the paper should be more appropriate for a specialized journal.

Specific issues:

It is not fully clear how amplification was quantified. From Supplemental Figure 5, it seems that amplification is not higher than 2-3-fold. However, in Figure 3 numbers are higher. Please clarify.

The amplification was quantified using two different techniques: qPCR of genomic DNA (Figure 3C), and coverage depth from whole genomic sequencing (Figure 3B, Supplementary Figure 5). Apart from the biases inherent in the two techniques (e.g. due to primer sequence, or read mapping), we believe the differences in the values obtained from the two techniques are because the coverage depth is normalized to the coverage across the entire genome in the strain, while the qPCR data is normalized to the *rpoC* gene, and then to the WT values. To test the possibility of differences in normalization explaining the apparent differences in amplification, we similarly normalized our coverage data to the coverage of the *rpoC* gene, and then to the WT (new Supplementary Figure 9) and see results similar to the qPCR data in Figure 3C.

Additionally, because the amplification is unstable, the copy number can vary several fold depending on the growth conditions (for example, as seen in Figure 4A, 4B). Further, the copy number is also likely heterogeneous within a strain (resulting in the heteroresistance seen in Fig 5C), and sampling differences within the small amounts of DNA used for the WGS library preparation and qPCR may also contribute to the variability seen in the copy number across experiments. High-depth long read whole genome sequencing would ideally capture the copy number distribution of a strain, and would enable more precise comparisons. We have added these details to the methods and main text:

Line 418-423: “We quantified the copy number of the amplification using both qPCR on genomic DNA, as well as coverage depth in our whole genome sequencing samples. However, given the amplification instability, and consequent heterogeneity even within an otherwise isogenic population of a single strain, high coverage long read sequencing

may be necessary to provide a more precise measurement of the copy number distribution of the amplification in a population.”

Line 533-536: “For the data shown in Supplementary Figure 9, we calculated the normalized coverage of *sdrM*, *lmrS*, *sepA*, and *rpoC* as the sum of the reads that mapped to the gene divided by the gene length. The efflux pump normalized coverage was then divided by that of *rpoC*, and for each evolved isolate, normalized to the WT value to get the fold-change in coverage.”

Also, in Figure 3, Copy number and level of expression do not always correlate: see for instance *lmrS* in 1.7a and 4.2a and the explanation of the high increase in expression with not so high increase in amplification is weak. Please discuss in more detail.

We appreciate the reviewer’s concern that the copy number of the amplification does not always correlate with the level of expression (as seen in Figs 3C and 3D). This suggests that the regulation of these genes is changed upon amplification. We hypothesize that this is due to read-through transcription from the highly expressed tRNA-rRNA gene cluster, that’s located upstream of the efflux pumps in the amplification. A similar phenomenon has been seen in *Streptococcus pneumoniae*. We have now expanded on this potential explanation in the manuscript:

Lines 335-343: “In the evolved isolates with amplifications, the expression levels of *sdrM* and the adjacent efflux pumps *lmrS* and *sepA* were much higher compared to the copy number of the coding genes (Figures 3C and 3D) suggesting that the regulation of these genes is altered in the amplification. Interestingly, a cluster of tRNA-rRNA genes, which are typically extremely highly expressed, is located downstream of *sdrM*, *lmrS*, and *sepA*, but gets positioned upstream of the amplified copies of the efflux pump encoding genes. This raises the possibility that read-through transcription from the tRNA-rRNA genes may result in the highly increased efflux pump expression we observed in our evolved isolates containing amplifications, similar to a previous observation in *Streptococcus pneumoniae*.”

I guess that the authors have measured MIC by double dilution. If this is true, values of 2-fold change must be taken with caution, because this is the error of the method. Further MIC means Minimal Inhibitory Concentration. This means that, if the experiment is repeated, the MIC is not the average. It is the minimal concentration that is obtained. If several values are high and one is low, it is acceptable to discard this one, but for some values are high and some are low, an average cannot be used.

For small changes in MIC as here reported, it is however acceptable to make growth curves in presence of different concentrations of antibiotics, use intermediate concentration or Epsilon tests.

We have calculated the MICs using a double dilution method to measure the optical densities at the different DLX dilutions, and then fitting the plot of these values against

the DLX concentrations to a modified Gompertz function (as detailed in Lambert et al 2001 – reference 81 in the new manuscript). This allows for a more exact determination of the MIC value. Additionally, we believe that variability in MIC measurements reflects the biological and technical variability/noise associated with the strains and the experimental set-up. We have therefore depicted all individual values for our MIC measurements, in addition to the mean and standard deviation, to show the full variability of the experiment, and allow for appropriate interpretation. We have added the MIC calculation details to the methods section:

Lines 631-634: “After growth, the OD600 of all wells was measured using a plate reader. The OD600 measurements for all DLX concentrations were plotted against the DLX concentration values and fitted to a modified Gompertz function to determine the exact MIC values.”

Reviewer #3 (Remarks to the Author):

The paper by Silva et al represents an interesting description of a novel mechanism of resistance to a 4th generation Fluoroquinolone Delafloxacin. The study appears to have been well executed, robust and with appropriate conclusions. The findings provide potentially important insights relevant to the development of antimicrobial resistance via conserved efflux pumps. I have some suggestions to improve the manuscript and to understand the broader clinical relevance of the discovery.

Major:

It would be extremely informative to examine the allelic diversity of the SdrM gene across the *S. aureus* species diversity to examine amino acid variation that may impact on function. For example, are the sdrM mutations identified via experimental evolution represented among clinical isolates ? Are other mutations identified that may affect the binding pocket of the efflux pump ? While I appreciate that the drug has not been in clinical use for long, it is feasible that resistant strains may have evolved already, and/or some strains may have sdrM mutations that may make them more susceptible to developing resistance.

We thank the reviewer for this interesting suggestion and have now examined the allelic diversity of SdrM. We downloaded the genomes of ~64,000 *S. aureus* isolates from the NCBI Pathogen Detection database (<https://www.ncbi.nlm.nih.gov/pathogens/>). We then performed a tblastn search against these sequences using the JE2 SdrM protein sequence as a query. We identified a large number of polymorphisms in SdrM, including in the binding pocket. While there were no mutations at position Y363, 8 strains had mutations at position A268, including one with an A268S mutation. As the reviewer points out, delafloxacin has not been in clinical use for too long, and the low frequency of A268 and Y363 mutations may reflect that. Further, it is possible that the other mutations seen in binding pocket residues may also alter resistance to delafloxacin, or possibly other antibiotics. These results are shown in the new Supplementary Table 3.

We have included a description of these results, as well as the associated methods in the manuscript:

Line 162-169 (Results): “We examined the allelic diversity of the SdrM protein to determine whether the evolved alleles we identified (A268S and Y363H), or any additional mutations in the binding pocket of SdrM, are seen in publicly available genomes of *S. aureus* isolates. The JE2 SdrM protein sequence was queried against a set of 63,980 *S. aureus* genomes from the NCBI Pathogen Detection database (Supplementary Table 3). While no mutations were seen at position Y363, we identified one strain with an A268S mutation, and a further seven strains with an A268V mutation. Additionally, these *S. aureus* strains harbored numerous mutations in other residues of the predicted SdrM binding pocket.”

Lines 396-400 (Discussion): “Our analysis of more than 63,000 *S. aureus* genomes also showed that A268S and Y363H mutations are rare. However, other mutations in SdrM, including in residues predicted to be in the binding pocket, are abundant, and these may potentially affect *S. aureus* resistance, and evolution of resistance, to DLX and other antibiotics.”

Lines 564-572 (Methods): “Analysis of allelic diversity of SdrM
Genome assemblies of 63,986 *S. aureus* isolates were downloaded from the NCBI Pathogen Detection database (<https://www.ncbi.nlm.nih.gov/pathogens/>) which was accessed on January 19th, 2023. The FASTA sequences were concatenated, and a local BLAST database was created with the NCBI BLAST makeblastdb command. The protein sequence of the JE2 SdrM sequence was used as a query for a tblastn (translated BLAST) search in the local database, and the output was parsed for all hits that had a percent identity > 85%, E value < 10^{-6} , and 100% coverage. The amino acid frequencies for all positions were quantified from the tblastn BTOP output and are shown in Supplementary Table 3.”

The experimental evolution experiments described were carried out with the USA300 strain, it would be helpful to know if the described phenomenon relating to sdrM mutations and amplifications leading to resistance also applies to other strains/lineages of *S. aureus*.

We agree that it would be very informative to know whether clinical *S. aureus* isolates could also adapt to delafloxacin via *sdrM* mutations and amplification. We therefore performed new DLX resistance evolutions, similar to our original evolution, of three populations each of two clinical isolates from two different cystic fibrosis patients – a CC8 ST8 MRSA isolate and a CC1 ST188 MSSA isolate. We identified *sdrM* mutations in multiple populations and amplifications in all our evolved populations, indicating that this phenomenon is a general one. We have added these data as a new figure (Figure 6), and described it in the Results, Discussion, and Methods.

Line 285-295 (Results): *sdrM* mutations and amplification are prevalent in clinical isolates evolved for DLX resistance

We tested two *S. aureus* clinical isolates for the incidence of *sdrM* mutations and amplification upon evolution of DLX resistance. The two *S. aureus* clinical isolates were originally isolated from two cystic fibrosis patients. CF001 is a MRSA strain of clonal complex 9 (CC8) sequence type 8 (ST8) and CF106 is a CC1 ST188 MSSA strain (Supplementary Table 5). While CF001 had an DLX MIC ~2x lower than JE2, the DLX MIC of CF106 was ~50x lower than JE2 (Figure 6A). We evolved three independent populations each of both CF001 and CF106 in increasing DLX concentrations, and sequenced populations from intermediate passages (Figure 6B). Two out of three populations of both CF001 and CF106 had the *sdrM2 allele, while all CF001 and CF106 evolved populations had *sdrM* genomic amplifications.**

Line 385-387 (Discussion): There is significant diversity among *S. aureus* strains that are isolated in the clinic, but our experiments show that upon DLX exposure, selection for *sdrM* genomic amplifications, and at least the *sdrM2* allele, is common even in clinical isolates from different clonal complexes.

Line 485-488 (Methods): Similarly, for the evolution of the clinical isolates, three independent populations of CF001 and CF106 were evolved in parallel starting at a DLX concentration of 0.1 µg/ml for CF001 and 0.002 µg/ml for CF106. The concentrations were increased 1.5-2-fold at each exposure.

Line 513-519 (Methods): Analysis of clinical isolate genomes

The genomes of the clinical isolates CF001 and CF106 were assembled using Unicycler v0.4.8 on PATRIC (now integrated into the Bacterial and Viral Bioinformatics Resource Center). The assembled contigs were then annotated using Prokka v1.14.5. These annotated genomes were used as a reference to identify mutations in the respective evolved populations using breseq.

MLST of CF001 and CF106 was done using the PubMLST database online (pubmlst.org), and the SCCmec typing was done using Staphopia-sccmec.

Line 537-542 (Methods): For the clinical isolates, as the genomes were assembled into contigs, we compared the coverage of *sdrM* to the contig that contains it. The coverage depth of each base pair in the contig was determined as above. The fold-change in *sdrM* coverage was determined by dividing the number of reads per base pair of *sdrM* by that of the contig. The criteria to identify an amplification were the same as above, except that the relative coverage of *sdrM* was compared to the value of the respective parental clinical isolate.

Minor:

Is the *sdrM*-Tn insertion stable? Has the genome sequence of the *sdrM*-Tn mutant been determined to account for possible spurious mutations that could impact function?

We appreciate the concern that the transposon insertion in the *sdrM*::Tn mutant may not be stable, or that the *sdrM*::Tn mutant may have additional mutations that may impact function. To test this, we sequenced the *sdrM*::Tn strain, and did not see any additional mutations in the strain. Further, the mariner-based transposon *bursa aurealis* that was used to make the *sdrM*::Tn mutant is an inactive transposon without the transposase enzyme, and is therefore expected to be stable. However, we checked the stability of the transposon in the *sdrM*::Tn mutant, and the populations from the terminal passages of the *sdrM*::Tn evolution, as well as WT as a control, by PCR using primers flanking the transposon insertion site (new Supplementary Figure 11C). Absence of the WT-sized band in the *sdrM*::Tn mutant, and the populations from the terminal passage of the evolution, indicates that the transposon was stable. We have added this information to manuscript:

Line 244-247: “The *sdrM*::Tn strain had no other mutations compared to the WT, showed similar growth to the WT in M63 (Supplementary Figure 11A, 11B) and had an MIC ~2-fold lower than the WT (Figure 5A), indicating that SdrM contributes to the intrinsic DLX resistance level of the WT MRSA strain.”

Line 249-251: “We verified that the transposon insertion was stable during the evolution in all three *sdrM*::Tn evolved populations (Supplementary Figure 11C)”

Fig 1 legend requires clarification of which isolates represent intermediate v terminal passages. Also indicate that the term 'mutations can refer to a SNP or an amplification.

We have modified the figure legend of Figure 1 to clarify this.

Fig 1 legend: “The presence of mutations in genes encoding DNA gyrase subunits (*gyrA*, *gyrB*) and DNA topoisomerase IV subunits (*parC*, *parE*), the three mutant alleles *sdrM1**, *sdrM2** and *sdrM3**, and a genomic amplification containing *sdrM*, are shown for populations from intermediate passages, as well as the three isolates from the terminal passage, for the ten independently evolved populations. Blue or yellow squares show the presence of the mutation (SNP or amplification) in a population or a terminal isolate, respectively. The terminal DLX concentrations represent the final concentrations of the evolution experiment (at which point the isolates were obtained).”

l.88 '3 individual isolates'- presumably of 3 sampled ie 100% ?

Yes, we sampled only three isolates from each population. We have now clarified this:

Line 90-92: “Three individual isolates from the terminal passage of each evolved population were tested for DLX resistance and all isolates showed DLX MICs ranging between ~2-33 µg/ml (Supplementary Figure 1A), confirming the evolution of high DLX resistance.”

Please make sure throughout the manuscript that the terms isolate, strain and population are used appropriately eg l.17- 'we observed that strains'...should this be isolates ?

We have made the appropriate changes throughout the revised manuscript.

REVIEWERS' COMMENTS

Reviewer #1 (Remarks to the Author):

The changes in the revised manuscript, which include new experiments and explanations, mostly address my previous points well. The manuscript is improved.

One small remaining issue is that the authors state in their response that they changed Figure 4E (cross-resistance to other antibiotics) to log scale. However, in the revised main manuscript, Figure 4E remains identical. It would be good to implement this change.

I still find the evolvability claim a bit too strong even though the experiments have been improved. Supplementary Figure 11D clearly shows that a similar level of resistance can be achieved without SdrM.

Reviewer #3 (Remarks to the Author):

The authors have addressed all comments satisfactorily.

We thank the reviewers for their comments and have responded to the specific concerns below in bold.

Reviewer #1 (Remarks to the Author):

The changes in the revised manuscript, which include new experiments and explanations, mostly address my previous points well. The manuscript is improved.

One small remaining issue is that the authors state in their response that they changed Figure 4E (cross-resistance to other antibiotics) to log scale. However, in the revised main manuscript, Figure 4E remains identical. It would be good to implement this change.

We apologize for this error – by mistake, we uploaded an older version of the figure while submitting the revised version of our manuscript. We have now included an updated version where Figure 4E is shown on a log scale.

I still find the evolvability claim a bit too strong even though the experiments have been improved. Supplementary Figure 11D clearly shows that a similar level of resistance can be achieved without SdrM.

We have qualified the evolvability claim to clarify that the frequency of resistance evolution is increased by the presence of SdrM, although the eventual levels of resistance achieved are similar in the presence or absence of SdrM.

Lines 356-358: “While the eventual level of DLX resistance attained was similar between the WT and *sdrM*::Tn strains (Supplementary Figure 11d), the presence of *sdrM* in the WT allowed for more rapid and frequent evolution of DLX resistance (Figure 5d).”

Reviewer #3 (Remarks to the Author):

The authors have addressed all comments satisfactorily.